



# Observed slump of sea land breeze in Brisbane under the effect of aerosols from remote transport during 2019 Australia mega fire events

Lixing Shen[1], Chuanfeng Zhao[1]*, Xingchuan Yang[1], Yikun Yang[1], Ping Zhou[1]

[1]College of Global Change and Earth System Science, and State Key Laboratory of Earth Surface Processes and Resource Ecology, Beijing Normal University, Beijing 100875, China

*Correspondence to:* Chuanfeng Zhao (czhao@bnu.edu.cn)

**Abstract.** The 2019 Australia mega fires were unprecedented considering its intensity and consistency. There have been many researches on the environmental and ecological effects of the mega fires, most of which focused on the effect of huge aerosol loadings and the ecological devastation. Sea land breeze (SLB) is a regional thermodynamic circulation closely related to coastal pollution dispersion yet few have looked into how it is influenced by different types of aerosols transported from either nearby or remote areas. Mega fires provide an optimal scenario of large aerosol loadings. Near the coastal site of Brisbane Archerfield during January in 2020 when mega fires were the strongest, reanalysis data from Modern-Era Retrospective analysis for Research and Applications version 2 (MERRA-2) showed that mega fires did release huge amounts of aerosols, making aerosol optical depth (AOD) of total aerosols, Black Carbon (BC) and Organic Carbon (OC) approximately 240%, 425%, 630% of the averages of other non-fire years. Using 20 years' wind observations of hourly time resolution from global observation network managed by National Oceanic and Atmospheric Administration (NOAA), we found that SLB day number during that month was only four, accounting for 33.3% of the multi-years' average. The land wind (LW) speed and sea wind (SW) speed also decreased by 22.3% and 14.8% compared with their averages respectively. Surprisingly, fire spot and fire radiative power (FRP) analysis showed that heating effect and aerosol emission of the nearby fire spots were not the main cause of local SLB anomaly while the remote transport of aerosols from the fire center was mainly responsible for the decrease of SW, which was partially offset by the heating effect of nearby fire spots and warming effect of long-range transported BC and $CO_2$. The large scale cooling effect of aerosols on sea surface temperature (SST) and the burst of BC contributed to the slump of LW. The remote transport of total aerosols was mainly caused by free diffusion while large scale wind field played a secondary role at 500 m. Large scale wind field played a more important role in aerosol transport at 3





km than at 500 m, especially for the gathered smoke, but free diffusion remained the major contributor.
The decrease of SLB speed boosted the local accumulation of aerosols, thus further made SLB speed
decrease, forming a positive feedback mechanism.
**1. Introduction**
Aerosols play an important role in balancing the Earth's radiation budget, through its direct or indirect
effect (Albrecht, 1989; Garrett and Zhao, 2006; IPCC, 2013; McCoy and Hartmann, 2015). There are
different kinds of aerosols from various sources which have different climatological forcing effects
(Charlson, 1992; Yang et al., 2016). Aerosols differ in radiative forcing effects as their physical and
chemical properties vary, some of which may affect the earth-atmosphere system by bringing changes
to the lifespan of clouds (Albrecht, 1989; Zhao and Garrett, 2015).
Carbonaceous aerosol contains black carbon (BC) and organic carbon (OC) and serves as a major
radiation-influencing aerosol which mainly comes from biomass burning (Vermote et al., 2009, Yang et
al., 2021). There have been studies addressing the importance of BC on atmospheric warming and that
of OC on weakening *in situ* downwelling solar radiation (Jacobson, 2001; Ramana et al., 2010). There
were also some studies trying to quantify the average radiative forcing of BC and OC while they
emphasized the potential uncertainties with respect to the specific values too (Zhang et al., 2017). At a
planetary scale, the change of aerosols brings many uncertainties to radiation balance thus further
influences the magnitude of atmospheric circulation (Wang et al., 2015; Zhao et al., 2020). At a
synoptic scale, aerosols can affect tropical cyclone by enlarging its rainfall areas which is also related
to its radiative properties (Zhao et al., 2018). At a regional scale, Han et al. (2020) discussed in detail
the radiative forcing effect of aerosols on the speed of Urban Heat Island (UHI) during different
seasons.
As mentioned above, biomass burning is an important source of aerosols, especially for carbonaceous
aerosols. Adequate amounts of fire-emitted aerosols would bring perturbations to the balanced Earth's
climate system through both direct and indirect effects (Jacobson, 2014). There have been many
researches discussing the characteristics of wild fire aerosols and their effect around the world
(Grandey et al., 2016; Mitchell et al., 2006). For example, Portin et al. (2012) investigated the
characterization of burning aerosols in Eastern Finland during Russian wild fires in summer 2010.



Kloss et al. (2014) pointed out wild fires could have plumes ascending high and polluting remote areas
with the help of monsoon. Grandey et al. (2016) quantified the total fire aerosol radiative effect over
the globe, which is estimated to be -1.0 W/m$^2$ on average. The fire aerosols could have more significant
radiative effects with clouds than under clear sky condition through cloud-aerosol interaction, whose
global forcing effect could reach -1.16 W/m$^2$ (Chuang et al., 2002).
Australia is one of the areas where wild fires occur frequently (Yang et al., 2021). There are nearly
550,000 km$^2$ of tropical and arid savannah burnt each year in Australia, contributing to about 6%–8%
of global carbon emissions from biomass burning (van der Werf et al., 2006; Meyer et al., 2008).
Particularly, there have been many studies concentrating on wild fires' association with enhancing
aerosol loadings and air pollution events in Australia, some of which included the discussion of
combined effect from background meteorological conditions (Mitchell et al., 2006; Luhar et al., 2008;
Meyer et al., 2008; Mitchell et al., 2013; Mallet et al., 2017). The 2019 Australia wild fires from
December 2019 to February 2020 were unprecedented in recent decades in terms of its magnitude and
consistency so that they have attracted the attention of the world in a short time. Numerous studies
have been carried out since their outbreak from different aspects. For example, Yang et al (2021)
examined the statistical properties of aerosol properties associated with 2019 Australia mega fire events
in both horizontal and vertical directions. Torres et al. (2020) investigated the aerosol emissions during
the mega fires happening in New South Wales, Australia and found a great amount of carbonaceous
aerosols in the stratosphere. Ohneiser et al. (2020) traced wildfire smoke in one of the most severe
burnt areas in southeastern Australia and found that smoke could even travel across the Pacific, which
was detected by an observation site at Punta Arenas in South America.
Sea land breeze (SLB) is a common circulation over coastal areas whose direct cause is the temperature
difference between land and sea (TDLS). Many studies have investigated this regional circulation. On
one hand, the complicated influencing factors of SLB have been studied from different perspectives
(Miller et al., 2013). Our previous studies pointed out that the change of TDLS is highly related to the
change of *in situ* downwelling solar radiation (Shen et al., 2021; Shen et al., 2021; Shen and Zhao,
2020). We also found that the continuous increase of surface roughness in cities could reduce the SLB
speed in long term (Shen et al., 2019). The long-term significance and trends of SLBs over the globe
are driven by climate regimes which are related to climatological differences in both *in situ*
downwelling solar radiation and background wind fields. There are also many other studies on the



influencing factors of SLB during short periods. For example, based on the case analyses, Sarker et al.
(1998) found that UHI magnitude has a great impact on the encroachment range of sea wind (SW)
frontal surface. Using regional model simulation, Ma et al. (2013) found that UHI effect could enhance
TDLS a lot which would result in strengthened SLB circulation in a great metropolis. Miller et al.
(2013) reviewed the SLB and pointed out that local topography such as the shape of the coastline, is
another important influencing factor of SLB. On the other hand, SLB's effect has also been extensively
investigated. For example, SLB has been reported as a direct controller of air pollutants which
transports air pollutants inland or to the vast ocean with the help of background meteorological field
(Nai et al., 2018; Shen and Zhao, 2020). SLB is also essential to modify the meteorological conditions
and local climate (Rajib and Heekwa, 2010). Moreover, SLB is a determinant factor of the diurnal
variation of the precipitation on the island since its direction and magnitude can affect the location and
magnitude of convective systems (Zhu et al., 2017).
Over the years, the cause and effect of aerosols, wild fires in typical areas, and SLBs have been learned
in detail respectively. The relationship between aerosols and other small scale circulations such as UHI
circulation has also been investigated from many aspects (Han et al. 2020). However, few studies have
investigated the effects of different types of aerosols on SLBs or looked into how local and remote
aerosol loadings during mega fires would affect local SLB with the help of meteorological background
field or other potential mechanisms. There was an updated and important study calling for attention of
the record-breaking aerosol loadings during 2019 Australia mega fires which led to cooling effect to
ocean temperature (Hirsch and Koren, 2021). Since *in situ* downwelling solar radiation and SST, which
are both important influential factors of SLB, are deeply affected by the radiative effects of different
types of aerosols, it is interesting to examine in detail how the record-breaking mega fires would
influence SLB by releasing large amounts of aerosols.
The paper is organized as follows. Section 2 describes the observation site, data and analysis methods.
Section 3 illustrates the characteristics of SLB, the variation of SLB days, the distribution and fire
radiative power (FRP) of wild fire spots, the anomaly of observed SW speed, land wind (LW) speed
and air temperature, the effects of different aerosols on SLB's variation, the analysis on background
wind field and the comparison between local fire spots' and the remote fire center's contributions.
Section 4 summarizes and discusses the findings of the study and proposes a mechanism of
aerosol-SLB interaction during the 2019 Australia mega fires' most intense period.



**2. Data and methods**
**2.1 Site**
The 2019 Australia mega fires occurred mainly in the eastern and southeastern coastal areas of
Australian continent (Yang et al., 2021). The southeastern parts, including the State of Victoria and
southeastern part of the State of New South Wales, belong to Marine Climate where obvious existence
of SLB (OE-SLB) is not clearly verified because of the influence of strong westerlies and water vapor
accompanied with westerlies from the ocean (Shen et al., 2021). Note that OE-SLB means that SLB is
significant from a climatological perspective. In other words, the SLB can be found during most time
of the year. Details about the definition of OE-SLB can be found in Shen et al. (2021) and are not
repeated here. Meanwhile, the wild fire events were most severe with a great density according to
numerous reports, which could possibly cause fire-induced complex flows and circulation in the form
of fire-atmosphere interactions in the vicinity of a fire (Stageberg, 2018). Based on previous
observation during mega fire events, the concentrated fire spots changed the local air pressure field and
added a regional temperature-pressure field, bringing uncertainties to local wind speed and wind
direction (Jia et al., 1987; Li et al., 2016). On one hand, this could further interrupt the SLB formation
since it might make the background wind field more complicated. On the other hand, the detected SLB
might not be accurate since it is likely to contain other wind disturbance at a small regional scale.
As shown in Fig. 1, we selected an urban site in Brisbane along the eastern coast of Australia as the
study site, which was due to several considerations. First, while the eastern coasts of Australia belong
to monsoon climate, the Australian monsoon system is not strong so that the OE-SLB can be verified
from a climatological perspective, which also means integrated SLB circulation can be found during all
seasons. Second, compared to rural sites, there are longer period of high time resolution observation
data at urban sites, which is necessary for the extraction of SLB signals. Third, the urban area of
Brisbane is relatively small and is not very far from vast areas of forests which provide stable
combustion environment, ensuring the persistent effect of wild fires when they occur. Fourth, the UHI
effect, which could possibly interrupt SLB and bring errors when calculating SLB magnitude, should
be small for the study region considering the small scale of urban areas. Also, the wild fires near
suburban areas could further eliminate the UHI effect even if it could exist through their heating impact
on these areas. In contrast, the forest site is surrounded by or within great amounts of flora where a




majority of solar radiation is absorbed and scattered by leaves, prohibiting the surface heating by solar
radiation and then the formation and detection of SLB. Actually, due to the existence of photosynthesis,
the heat absorption process of leaves from solar radiation and the temperature rise of 'leave surface' are
different from those of Earth surface. As a result, the traditional mechanism of SLB formation is not
necessarily applicable when the site is in the forest or quite close to clusters of flora. Considering all of
these, we chose the site of Brisbane Archerfield located at eastern coast of Queensland State (Fig. 1) as
the study site.
**2.2 Data**
The Several types of data have been used in this study, including the land cover type data, the
Modern-Era Retrospective analysis for Research and Applications version 2 (MERRA-2) data, the
Moderate Resolution Imaging Spectroradiometer (MODIS) data, the ground site observation data, the
Fifth Version of European Centre for Medium-Range Weather Forecasts (ECMWF) ReAnalysis (ERA5)
data, the Firespot and FRP data, and the Global Data Assimilation System (GADS) data. The detailed
data information is described below one by one.
Land cover type data: The land cover type data of Australia is from Dynamic Land Cover Dataset
(DLCD) with Version 2.1 provided by Geoscience Australia. In this study, the DLCD land cover type
data was used to reveal the surrounding landscape of Brisbane Archerfield. The spatial resolution of the
data is '0.002349°×0.002349°', which is based on the annual mean of data from 2014 to 2015.
MERRA-2 data: MERRA-2 belongs to the global atmospheric reanalysis product managed by National
Aeronautics and Space Administration (NASA). It is produced by the Global Modeling and
Assimilation Office (GMAO) and the assimilation system of Goddard Earth Observing System
(GEOS-5) is used to ensure the quality of this dataset. For major ground sites over Australia, Yang et al.
(2021) compared its monthly aerosol optical depth (AOD) product with Aerosol Robotic Network
(AERONET) observations and found their RMSEs were all smaller than 0.05. Thus, MERRA-2 should
be reliable to be used for the analysis of the large-scale spatial distribution of AOD in Australia. Yang et
al. (2021) also denoted that the 2019 Australia mega fires were the strongest in January of 2020.
Correspondingly, we used the monthly AOD of January at 550 nm from 2002 to 2020 to check the
AOD difference between the mega fire year and years with no mega fires. The spatial resolution of
MERRA-2 AOD data is '0.625°×1°'.



MODIS data: The MODIS instrument is performed on Aqua and Terra platforms. In this study, we used
the MODIS cloud product which belongs to the dataset of MCD06COSP_M3_MODIS. The cloud
information includes cloud optical depth (COD) and cloud fraction for all January months during the
period from 2003 to 2020 with monthly time resolution. The Brisbane Archerfield site is located at
'153.008°E, 27.57°S'. So we used COD and cloud fraction data whose space range and resolution are
'152.5°E-153.5°E × 28.5°S-26.5°S' and '1°×1°' respectively. This space range covers the whole
Brisbane area and the normal encroaching distances of SLB which are about dozens of kilometers
(Rajib and Heekwa, 2010; Shen et al., 2019). In this study, the spatial averages of them were calculated
to represent the local COD and cloud fraction during every January from 2003 to 2020.
Ground site observation data: The wind and air temperature observation data are from National
Oceanic and Atmospheric Administration (NOAA) global observation network at the site of Brisbane
Archerfield (153.008°E, 27.57°S). We used data in January from 2001 to 2020 in this study. The time
resolution is every 3 hours at 200, 500, 800, 1100, 1400, 1700, 2000, 2300 UTC. The wind information
includes wind speed and direction with few missing observations. The air temperature is measured in
Fahrenheit and we have converted it into Celsius. The observation data was the main data used in this
study to show the variations of both SLB and temperature during the fire.
ERA5 data: The monthly mean Uwind (zonal) and Vwind (meridional) of January 2020 from the
ERA5 were used in this study to reveal the background meteorological field so as to assess its effect on
aerosol transport. The spatial resolution is '0.25°×0.25°' at pressure levels of 1000 hPa, 975 hPa, 950
hPa, 925 hPa, 900 hPa, 875 hPa, 850 hPa, 825 hPa, 800 hPa, 775 hPa, 750 hPa and 700 hPa.
Firespot and FRP data: Firespot and FRP data are from MODIS product (MCD14). It can catch and
locate the active fire hotspots based on thermal anomalies of 1 km pixel resolution (Giglio et al., 2016).
The time resolution is daily and we used the monthly averages for January from 2002 to 2020 to look
into the fire situations over the years in detail.
GDAS data: The GADS data was used to perform the back-trajectory analysis from the Hybrid
Single-Particle Lagrangian Integrated Trajectory (HYSPLIT). The spatial resolution of GADS data is
'1°×1°' with daily time resolution. The levels of GDADS data chosen in this study to help to perform
HYSPLIT analysis were 500 m and 3 km respectively. The time range set in this study was the January
of 2020.





**2.3 Methods**
**2.3.1 Extracting SLB signal**
The verification of OE-SLB and extracting of SLB signals from original wind observation over
monsoon areas were carried out through the method of Separation of Regional Wind Field (SRWF).
The definition of OE-SLB, the details of SRWF method and criterion for verification were detailed in
our previous studies and not repeated here (Shen et al., 2019; Shen and Zhao, 2020; Shen et al., 2021).
Briefly speaking, SRWF calculates the vector difference between observed wind vector and daily
average wind vector for each observation time. Then, the vector difference is considered as the local
wind. The criterion of OE-SLB requires that there exist intersection sets among the range of SW, the
range of LW and the range of hourly average of wind angle in a diurnal period (HAWADP). Also, the
intersection set between the range of SW (LW) and the range of HAWADP only exists during daytime
(nighttime). Then the local wind can be thought as the SLB signal as long as the OE-SLB is verified at
that site. Based on HAWADP and specific sea-land distribution, we further defined the prevailing time
of sea wind (PTS) and prevailing time of land wind (PTL). Briefly speaking, during PTS (PTL) the
local wind keeps blowing from sea (land) and the wind angle keeps rotating towards the direction of
vast sea (inland). The HAWADP at Brisbane Archerfield is shown in Fig. 2. As shown, the HAWADP
of local wind was close to sinusoid, which conformed to previous findings in other monsoon areas
(Shen et al., 2021; Yan and Anthes, 1987). According to the sea-land distribution shown in Fig. 1, we
first defined the ranges of SW and LW and then the OE-SLB of Brisbane Archerfield was verified
using these criteria. We further selected the PTS (PTL) based on the rules above.
To make it clear, we summarized the range of SW, LW, PTS and PTL in Table 1. Note that the actual
PTS (PTL) may be longer than what we defined here because the time resolution is 3 hours instead of
hourly in this study. As a result, we cannot know the exact threshold of time when the wind angle meets
the criteria mentioned above. For instance, it is possible that the wind angle is within the range of SW
before 0500 UTC. However, it is still sure that the SW (LW) develops vigorously during 0500-0800
UTC (1400-2000 UTC) based on Fig. 2, which means that '0500-0800 UTC' and '1400-2000 UTC' are
within the real PTS and PTL respectively even if they are not the exact PTS or PTL. Thus, the defined
PTS (PTL) in this study is reliable. The aim to define PTS (PTL) is to find the time period when SW
(LW) develops most vigorously so as to ensure further exclusion of winds from synoptic scales when





extracting real SLB signals after applying the SRWF method (Shen and Zhao, 2020; Shen et al., 2021;
Cuxart et al., 2014).

### 2.3.2 Definition of the SLB day

SLB day is the day when SLB circulation is most significant (Xue et al., 1995). To some extent, the
number of SLB days reveals the activity level of SLB. Different criteria have been adopted when
defining SLB day. Here we referred to our previous study (Shen et al., 2019) to adopt the criteria based
on the minimum times of successful detection of winds coming from the range of SW (LW) during
PTS (PTL). Since the time interval between two adjacent observations is 3 hours, which makes the
total observation hours less than the total hours during prevailing time, we modified the criteria slightly
as follows: when the offshore land winds occur in the time period of 1400-2000 UTC with total
occurrence time no less than 3, and the onshore sea winds occur in the time period of 500-800 UTC
with total occurrence time no less than 2, the day is counted as a SLB day.

### 2.3.3 The calculation of monthly SW and LW speeds

After defining PTS, PTL and SLB day, we could finally calculate the monthly SW and LW speeds.
First, we picked up SLB days in every January from 2001 to 2020. Second, we picked up local wind
speed during PTS (PTL) on SLB days and calculated the monthly average of SW (LW) speed in every
January from 2001 to 2020.
Based on GDAS data throughout the whole January in 2020, the back trajectories of lower atmosphere
at Brisbane Archerfield were simulated using the HYSPLIT model. This could help analyze the
transport effect of background wind fields on aerosols at this site. The simulated levels at the site were
500 m and 3 km since the lower level of atmosphere (500m) was closer to fire spots and there was also
accumulated smoke at 3 km in the southeastern parts of Australia during the exact same month (Yang et
al., 2021). The TrajStat module of Meteoinfo version 2.4.1 was also used to cluster the back trajectories
based on the Euclidean distance method, whose details and source code could be found at its official
website (http://meteothink.org/docs/trajstat/index.html, last access: 31 January 2021).

### 2.3.4 The calculation of monthly temperature during daytime and nighttime

After defining the SLB day, PTS and PTL, we calculated the monthly temperature during daytime and
nighttime using the similar method as SW and LW speeds. First we selected the temperature on SLB





days. Second, we calculated the monthly average of temperature during PTS (PTL) to represent
monthly average temperature during daytime (nighttime) in January. Actually, temperature during
daytime (nighttime) represents land temperature when SW (LW) prevails. In order to make it clear and
concise, we call it temperature during PTS (PTL) or land temperature during daytime (nighttime) in
this study.
**3. Results**
**3.1 The variation of SLB day number**
Figure 3 shows the SLB day number in January from 2001 to 2020. As shown, the SLB day number in
January was normally larger than 10. Among these 20 years, there were 25% of the years whose SLB
days in January accounted for more than half of the month. Note that it does not necessarily mean that
there is no SLB on days that are not SLB days. It is obvious that there was a slump in the number of
SLB day in 2020. The total SLB day number dropped to only 4 during the mega fire events, accounting
for only 33.33% of the average SLB day number during the past 20 years. Also, year 2012 also
witnessed low SLB day number (6 days) in January. There are a lot of potential influencing factors for
SLB frequency, such as the background wind field (Miller et al., 2013) and the interruption of other
small scale circulations (Kusaka et al., 2000). Among all the influencing factors, cloud is one of the
most important because it has significant effect on *in situ* solar radiation which is the direct cause of
TDLS. We would discuss this in the following sections.
**3.2 The trends in SW and LW speeds and local air temperature**
The monthly mean SW and LW speeds in January are shown in Figure 4a. As can be seen, there were
fluctuations of both SW and LW speeds in January from 2001 to 2020. The SW speed was higher than
LW speed, which conformed to many previous findings (Miller et al., 2013; Zhu et al., 2017). The
averages were calculated as 3.70 m/s for SW speed and 2.86 m/s for LW speed, respecitvely. Figure 4b
and c show the anomalies for both SW and LW speeds. In general, LW speed fluctuated more
significantly than SW speed did. This is due to its lower level of kinetic energy which could make it
more sensitive to any potential interruptions from the background meteorological field (Shen and Zhao,
2020). The negative anomalies of LW speed happened in 2001, 2004, 2008, 2010, 2011, 2015, 2016,
2017, 2018 and 2020. Different from other years, it is obvious that the negative anomaly in 2020 was





higher than 0.6 m/s, which was beyond the multi-year oscillation range. The anomaly accounted for
22.3% of multi-years' average LW speed. The negative anomalies of SW speed happened in 2004, 2008,
2009, 2010, 2011, 2013, 2014, 2015, 2017 and 2020 (Figure 4c). For SW speed, the negative anomaly
value in 2020 was also obvious but it was still within the multi-year oscillation range. It was higher
than 0.5 m/s, accounting for 14.8% of the multi-years' average. It is interesting to find that there were
obvious positive anomalies of both SW and LW speeds in 2003 whereas their absolute values were not
the highest. Also, the SLB day number in 2003 was near the average. We will further discuss this along
with the aerosol emissions during that year in the following sections.
It can be seen in Figure 4b that there were also significant fluctuations in nighttime land temperature
over the years. There was a soar in land temperature in 2020 which approached nearly 24 °C. It was
nearly 3 °C more than the multi-years' average, exceeding the range of multi-years' oscillation. The
fluctuation of land temperature during daytime was less significant than that during nighttime. There
was obvious positive anomaly in 2020, indicating that the daytime land temperature was higher than
those in normal years. Meanwhile, it was still within the range of multi-years' oscillation range though
the positive anomaly was obvious. Fire spots have heating effect on the nearby environment through
either shortwave radiation of light from fires or heat conduction caused by temperature gradient. It can
be inferred that mega wild fires during January 2020 contributed to the positive temperature anomalies
during PTS (PTL) through the heating effect of fires though it might not be the only cause. The heating
effect during fire events was more significant during nighttime than daytime. This is probably due to
colder background temperature field during nighttime.
Basically, the decreased SW (LW) speed revealed that the TDLS during PTS (PTL) decreased. To be
more specific, the temperature difference between the small regions where the upward stream and
downward stream of SLB circulation lie respectively became smaller during January 2020. Based on
Figure 4b and c, temperature during PTL seems to be negatively related to LW speed anomaly while it
is obvious that temperature during PTS does not show any corresponding relationship with SW
anomaly.
In order to be more accurate, we carried out linear regression between temperature during PTL and LW
anomaly and found that they had negative linear relationship ($p < 0.02$) with each other (Figure 5). As
the temperature increased by 10 °C, the LW speed anomaly decreased by 1.52 m/s. During nighttime,
the land is colder than the sea. As the land temperature increases, the TDLS becomes smaller if the SST



where the upward stream of SLB lies remains relatively stable, so does the LW speed. Shortly, the good
linear relationship reveals that the variation of temperature during PTL (nighttime land temperature)
could generally represent the variation of TDLS during PTL while the daytime land temperature
variation could not represent the TDLS variation during PTS. In our previous study, we also found
through observation that the daily lowest temperature (DLT) was well negatively related to LW speed
while the SW speed was more related to in situ solar radiation rather than merely land temperature
(Shen et al., 2021), which was similar to the findings here. It could be inferred that although the land
temperature during daytime increased during mega fire events, TDLS was still narrowed during fire
events. If we only consider the land temperature, the SW speed should have increased during fire
events because SW circulation is formed due to warmer land and colder sea. Consequently, there
should be other factors which could cause decreased TDLS during PTS, which is the direct cause of
SW speed decrease. We would investigate this in the following sections.
**3.3 The distribution and FRP of fire spots**
Since the heating effect depends largely on the distance between the area heated and the heat center, it
is necessary to examine the distribution of fire spots in January over the years, which is shown in
Figure 6. It can be seen that fire spots scattered all over the eastern part of Australia over the years
during January. January is the middle of Australian summer which is the season when wild fires are the
most frequent (Yang et al., 2021). Apart from 2020, other years also witnessed considerable scattered
fire spots all over the coastal and inland regions. It is obvious that there was an extreme fire center in
the southeastern corner of Australia with great density of fire spots in January 2020. This was exactly
the region where the 2019 Australia mega fires mainly happened. To be specific, it was the eastern
corner of Victoria State and the southeastern corner of New South Wales State, which conformed to
many reports from media. There was also a great fire center in the southeastern corner in 2003 although
the scale was smaller than that in 2020. Considering the distribution of fire spots near the site, the
density of fire spots nearby was not higher than in other years. Instead, there seems to be more fire
spots nearby the site in 2003, 2005, 2006, 2010 and 2013 in the figure. If we restrained the nearby
region to areas with smaller scale, year 2003 and 2013 rather than 2020 had the most nearby fire spots.
There exists another possibility that although the fire spots nearby the site were not more concentrated
with great density in 2020 than in other years, the FRP of fire spots in 2020 was higher. This means that



the fire was greater regardless of the ordinary density of spots, which could result in more fire aerosol
emissions. So we further examined the FRP of fire spots in 2020 and in other years. In order to make it
comparable and verifiable, the time period of data chosen here was the same as that in Figure 6. As
shown in Figure 7a, both the nearby and local fire spots in 2020 were mostly within the lowest FRP
range, which was less than 235 MW. There were some sparse fire spots with greater FRP (235-863
MW) scattered all over the eastern part of Australia. The FRP of the fire center was higher than the FRP
of other fire spots where there were many fire spots with greater FRP which belonged to the range of
'235-863 MW' or '863-2194 MW'. Figure 7b shows the FRP of all fire spots from 2002-2019. The
FRP of nearby or local fire spots were also with the lowest values. With the year number increased, the
density of fire spots with higher FRP (235-863 MW) increased significantly, most of which were
located at inland areas of Australia continent. This indicates that scattered wild fires with low or
medium FRP are common in Australia but concentrated mega fires are not so common. There were also
some fire spots which belonged to the range of '235-863 MW' or '863-2194 MW' in 2003, yet the
number was less and the distribution areas were smaller. Based on Figure 7, one important point we
found is that there was no discrepancy between FRP of nearby or local fire spots in 2020 and that of
nearby or local fire spots in other years. So the possibility mentioned above was discarded.
Based on the analysis above, the nearby fire spot density and FRP in 2020 were both at the same level
as other years for local regions near the site. This implies that the heating effect of nearby fire spots did
exist in 2020, contributing to the increase of land temperature to some extent, but it was not likely the
major cause of land temperature anomaly. Fluctuation of land temperature might be caused by
combined mechanisms or some other potential factors. In other words, the heating effect of fire spots
does not necessary correspond to observed air temperature increase. For example, Figure 4b and c
show that there were negative land temperature anomalies in 2003 but actually this year witnessed
greater density of nearby or local fire spots. In real situation, the scale of SLB is quite small. The fire
spots might be quite a long distance away from the area where vertical stream of SLB lies as a result of
which the heating effect is weak.
**3.4 The spatial distribution of aerosols**
Large fires would have great aerosol loadings which would affect the *in situ* solar radiation and then
the radiation budget. Based on the basic physical mechanism of SLB formation, the observed decreased



SW and LW speeds demonstrated the decreased TDLS. As mentioned above, the heating effect of
nearby fire spots was weak and did not become more significant in 2020. So the more important factors
bringing about the decrease of SW and LW speeds should be closely related to TDLS rather than the
land temperature only. The TDLS during SLB formation is highly related to the *in situ* downwelling
solar radiation. As the shortwave radiation increases, the TDLS becomes larger due to the different heat
capacities of land and sea. SW forms and prevails when TDLS is enough to drive the thermodynamic
circulation. During nighttime, the land-sea system is the heater for upper atmosphere as they both gives
out heat and undergoes energy loss in the form of longwave radiation. As the outgoing longwave
radiation increases, the TDLS also becomes larger due to the different heat capacities of land and sea.
Then the LW forms in the similar way as SW forms.
Based on discussions above, *in situ* downwelling solar radiation is a crucial factor for SW speed.
Considering that aerosol is an important factor affecting *in situ* downwelling solar radiation, it is
necessary for us to check the temporal and spatial variations of aerosols over the years. Figure 8 shows
the spatial distribution of AOD of total aerosols (TA-AOD) over the years using MERRA-2 aerosol
product. Over the years, the background level of TA-AOD was generally low in Australia, implying
that Australia was less polluted from human pollution. The TA-AOD in 2020 increased significantly
compared with the average level. It can be seen that there was a maximum value center in the southeast
corner, which overlapped the region of fire spots center (Figure 6). The peripheral area of maximum
value center was covered with isopleth showing the characteristics of free diffusion of aerosols in the
air. There was also a maximum value center in 2003 whose scale was smaller, overlapping the smaller
region of fire center in 2003. Based on findings from these three aspects, it can be concluded that the
mega fire center was the main source of large aerosol amounts around the site location. In general, the
TA-AOD was about 240% of the multi-years' average level at the site, while the TA-AOD in the fire
center was at a more astonishing level, accounting for more than 420% of the multi-years' average
level at the site. Aerosol could significantly affect the *in situ* downwelling solar radiation through direct
radiative forcing. Turnock et al. (2015) calculated the relationship between AOD and surface solar
radiation (SSR) and found that when the background value is low over the years, the SSR increases by
10% as AOD varies from 0.32 to 0.16. In this study, the TA-AOD increased even more significantly
(240%) considering the low background value. Normally, when we talk about the radiative forcing of
aerosols in the form of SSR difference, it means instantaneous radiative forcing. However, the



formation of SLB is the result of different levels of radiation accumulations between land and sea. So
the effect of aerosols on the total *in situ* downwelling solar radiation can accumulate in the process of
SLB formation and results in even more significant impacts on the change of surface temperature.
Apart from aerosols, clouds could play an even more important role in the radiation budget. The COD
and cloud fraction anomaly at this site are shown in Figure 9. The time range was from 2003 to 2020
due to data availability. It can be seen that both the cloud fraction and COD in 2003 were at an obvious
low level, while both the cloud fraction and COD in 2020 showed a tiny negative anomaly. Based on
the spatial distribution of TA-AOD, both 2003 and 2020 witnessed a soar in TA-AOD at the site while
TA-AOD increased more significantly in 2020. Figure 3 shows that there was a slump in SLB number
in 2020 while not in 2003, while Figure 4 shows that there were positive anomalies of both SW and
LW speeds in 2003. Many previous studies on SLB have pointed out that high level of *in situ*
downwelling solar radiation is favorable for SLB formation and SLB speed increase (Shen and Zhao,
2020; Shen et al., 2021, Miller et al., 2013). Our previous study in monsoon climate region also showed
that there was a positive linear relationship between *in situ* downwelling solar radiation and SW speed
(Shen and Zhao, 2020). As known, the *in situ* downwelling solar radiation is determined by both cloud
and aerosols through their combined 'Umbrella Effect'. The finding shown in Figures 3 and 4 could be
explained by the radiative cooling effects of aerosols and clouds. Although there was positive anomaly
of TA-AOD in 2003, the COD and cloud fraction was less than the average, offsetting the aerosols'
negative radiative forcing effect. *In situ* downwelling solar radiation of the regional sea-land system
was still ensured so that the SLB happened with a normal frequency (Figure 3) and with an even larger
speed (Figure 4). The *in situ* downwelling solar radiation in January 2020 should be lower than the
average, considering the tiny negative anomaly in both COD and cloud fraction and the significant
increase in TA-AOD. The increased radiative forcing effect of TA-AOD was accumulated during the
formation of SW. In conclusion, during daytime, the negative radiative forcing effect of total aerosols
was the determinant factor to weaken the *in situ* downwelling solar radiation, resulting in lower level of
TDLS and then decreased SW speed.
Mega fire events are special in emitting large amounts of carbonaceous aerosols which include OC and
BC. The OC is a very good scatter to solar radiation. Thus, among all the aerosols, OC could be an
important contributor of the weakened TDLS during SW formation. Figure 10 shows the spatial
distribution of OC over the years. The spatial distribution of OC was also similar as the fire spot



distribution, which further confirmed that the source of great aerosol emissions was from mega fire
events. There were extreme value centers in the fire center in both 2003 and 2020. Same as found
earlier, it can be seen that the large value spread to farther place in 2020 than 2003, indicating that the
fire events were more severe in 2020 than 2003. Similarly, the background value of OC at the site was
low on average. The specific value of OC-AOD at Brisbane site in 2020 was about 630% of the
multi-years' average, which was even higher than that of total aerosol. This is easy to understand
because the fire center is also covered with plants and trees and the combustion of them can bring
significant amounts of carbonaceous aerosols. Zhang et al. (2017) estimated the radiative forcing of OC
globally using BCC_AGCM2.0_CUACE/Aero model, which showed that Brisbane was within the
large value area with high levels of negative radiative forcing at the top of atmosphere. They also owed
this to biomass combustion. Thus, both total aerosol and OC made great contributions to SW speed
decrease by decreasing *in situ* downwelling solar radiation in January 2020.
The result above is analyzed based on the impacts of aerosols on solar radiation. However there is
almost no shortwave radiation during nighttime. Then one question pops up: why was the slump of LW
speed more significant, which indicated that the TDLS was significantly weakened at night in January
2020? While the heating effect of fire spots on nighttime land temperature did exist which was more
significant than that during daytime, it was not likely the main cause of weakened TDLS based on FRP
and fire spot distribution analysis. We next checked the spatial distribution of BC over the years in
Figure 11. It shows that BC-AOD at the site was about 425% of the multi-years' average level with the
extreme value center overlapping the area of that of fire spots density. Similar as the distribution of
TA-AOD and OC-AOD, the peripheral areas of maximum value center are covered with isopleth
showing the characteristics of free diffusion. BC is well known as a kind of absorbing aerosols which is
reported to have wider range of absorbing band than greenhouse gases, which can absorb broadband
radiation from visible light to infrared wavelength (Zhang et al., 2017). During daytime, it can absorb
solar radiation, longwave radiation from the warmer land, and shortwave radiation from local fires.
During nighttime, it has a warming effect on both atmosphere and Earth surface through longwave
radiation. As a result, it has a warming effect on the Earth-atmosphere system including the surface of
the regional land-sea system so that there was a temperature soar shown in Figure 4b. The soaring BC
during the mega fire heated the local atmosphere, which was like adding a 'heater' in the air. The
'heater' then gave out downward longwave radiation to the regional land-sea system. Just like the sun



during daytime, this could trigger a SW circulation anomaly, weakening LW circulation. Considering
the BC burst during mega fire events, it is nothing weird about its dominant role in local land
temperature increase. The mechanism proposed above can be summarized as follows. During nighttime,
the formation of LW originates from the process of heat release of both land and sea. As they both lose
heat with different paces due to different heat capacities, the TDLS is enlarged. During the mega fire
event, the upper atmosphere of the regional land-sea system is heated so that the vertical temperature
gradient is weakened, which is unfavorable for heat release of both sea and land surfaces. As a result,
the TDLS is significantly weakened.
Another potential contributing accelerator is $CO_2$ which is also the product of fires due to the
combustion of plants and trees. $CO_2$ is a kind of greenhouse gas which is likely to be engaged in the
same mechanism as BC to reduce TDLS during nighttime except that $CO_2$ cannot affect the
downwelling solar radiation. Details about this is not repeated again. However we should note that the
effect of $CO_2$ is based on theoretical analysis rather than observational verification due to the lack of
accurate observation data. Both BC and $CO_2$ reduce TDLS, which partially offset the enhanced
radiative forcing effect of total aerosols, but their combined warming effect is more significant during
nighttime than during daytime. That is most likely the reason (at least partially) that SW speed had
negative anomaly but was less significant than LW speed.
What we discussed above are all factors whose influences were restrained to a small scale. Although
SLB is a small scale system, it can still be affected by the variations of signals in a large scale, since the
local temperature is affected by both regional forcing and the variation of large scale background
temperature field. In our previous study, we weighed their contributions qualitatively (Shen et al.,
2019). We here simply discuss the potential change in large scale SST. Hirsch and Koren (2021)
warned the effect of record-breaking aerosol emission from this mega fire on cooling the oceanic areas.
On a large scale, its average radiative forcing on sea surface was -1.0 ± 0.6 W/m². The temperature
decrease of large scale sea surface could have negative forcing on the SST at a regional scale, though
the specific temperature variation of the sea surface where the SLB vertical stream lies might not be the
same.
We summarized all the influencing factors of TDLS at both regional and large scales in Table 2. Among
all these factors, aerosols, BC, OC and $CO_2$ had direct forcing on TDLS by changing the solar radiation
reaching the regional sea-land system. In contrast, heating effect of fire spots and large scale SST



signal had forcing on land temperature and regional SST respectively thus further had different forcing
effects on TDLS during daytime and nighttime. During 2019 Australia mega fires, TDLS during
daytime and nighttime both decreased under their combined forcing effects, which could be inferred
from the anomalies of SLB speed. Clearly, the directions of all forcing effects from different factors
were the same during nighttime. That was why LW speed decreased much more significantly than SW
speed did. The negative radiative forcing of total aerosols was the determinant cause for TDLS
decrease during daytime, which could only be partially offset by other factors.
**3.5 Source of aerosols**
**3.5.1 Fire center's emission**
As indicated earlier, year 2020 did not have advantages over other years in terms of local and nearby
fire spot density and FRP during January. Note that certain land cover type could also increase the
aerosol emissions. For example, if there were more combustible such as forests or plants, the fires
could emit more carbonaceous aerosols in form of smoke. Considering this possibility, we further
checked     the     latest     version     of     land     cover     in     Australia     online
(http://maps.elie.ucl.ac.be/CCI/viewer/index.php). It was updated to 2019 which overlapped with the
starting time of 2019 Australia mega fire events. It showed that the areas and density of flora near the
site were stable over the years, implying that the soar in local aerosols during mega fire events was not
likely caused by the change of land cover either.
As Figures 6-8 and 10 show, the distributions of fire spots, TA-AOD, BC-AOD and OC-AOD were
quite similar as each other. In the fire center, both the density and FRP of fire spots were much higher
in January 2020 than in January of other years. These are all based on distribution characteristics at a
large scale. In order to show the fire situation at the fire center more accurately, we magnified the FRP
map to restrain the areas to merely the fire center, which is shown in Figure 12. As shown, the fire spot
density was quite high in this area, especially along coastal areas. Compared with other areas, the fire
center had much more fire spots with higher FRP. The spots with FRP from 235 to 864 MW were
evenly distributed in all fired areas, surrounded by low FRP spots with high density. There were quite a
few spots with even higher FRP ranging from 864 to 2,194 MW, which could not be found in other
periphery areas (Figure 7a). In some areas at the fire center, we could even find fire spots with FRP
ranging from 2,194 to 5,232 MW. All these distribution characteristics of fire spots suggest the





possibility of large amounts of aerosols including smoke being emitted to the atmosphere, after which a
great concentration gradient in the horizontal direction formed between the fire center and farther areas.
Based on the basic Chemistry law, irreversible free diffusion would happen in this process. As the
concentration gap increases, the diffusion efficiency also increases. The distribution of contour lines in
Figures 8, 10 and 11 also shows the characteristics of free diffusion. Similar mechanism works out for
the spatial distribution of $CO_2$ during the fire events.
**3.5.2 Analysis on the background wind field**
Apart from free diffusion, wind is crucial for pollution transport including aerosols (Walcek, 2002).
Also, wind is a key factor influencing the near-surface $CO_2$ distribution (Cao et al., 2017). Zhang et al.
(2017) confirmed that BC could be transported through a long distance in mid-latitude areas. The
transport distance of OC was even longer than that of BC. It is necessary for us to look into the
background wind field in order to know the likely aerosol transport from the fire center to the site.
Yang et al. (2021) retrieved the average status of the vertical distribution of various aerosols in
southeastern Australia during 2019 Australia mega fires and found most of them accumulated under 3
km, which is about 700 hPa. Figure 13 shows the monthly average of background wind field based on
wind information at pressure levels from 1000 hPa to 700 hPa during January in 2020. The red cross
symbols represent the fire spot in this figure. The average background wind field clearly revealed the
existence of southern hemisphere's westerlies and subtropical high. The fire center was approximately
located at the intersection of the northern boundary of westerlies and southwestern boundary of
subtropical high. Since January is the middle month of Australian summer, the subtropical high
developed quite vigorously, some of which stretched into the eastern part of Australian continent. It
covered the areas where most fire spots were located. At a large scale, this brought quite hot and dry
background meteorological field, which was favorable for the development and persistence of wild
fires. Based on the average status of wind fields at different pressure levels, the subtropical high and
westerlies together formed a background wind field blowing from the site to fire center, which was not
favorable for the aerosol transport from the fire center to the site. However, we should notice that this
figure merely describes the monthly average status but ignores the status of wind flows at a more
accurate fine time scale. In other words, it is still possible that aerosols from the fire center were
transported to the site within some short periods in January 2020, thus made contribution to the positive



aerosol anomaly shown in Figures 8, 10 and 11. Based on the specific dates of SLB day during mega
fires identified in previous section, which were 4th, 14th, 20th and 28th in January respectively, we
divided the January in 2020 into five short time periods by excluding the identified SLB days. These
five short time periods were all named as 'No-SLB period'. We did the backward trajectory analysis
during each No-SLB period to see if the aerosols from the fire center were transported to the site with
the help of background wind field, thus further made this period a 'No-SLB period' through all the
mechanisms mentioned above. It is easy to understand that the near surface concentration of aerosol
should be at a high level in general not only because it was near the fire spots but also because it is
within boundary layer. Considering these aspects, the backward trajectory analysis was carried out at
500 m over the site. Figures 14a-e show the wind backward trajectories at this site during the five
No-SLB periods respectively. During the No-SLB periods of a, c, d and e, the winds mainly came from
the southern Pacific to the east of Australia continent, which could not transport aerosols from the fire
center. There were winds coming from the fire center merely during period b. The northern edge of
wind flow beam was quite near the fire center, then it went further towards the northeastern direction in
the southern Pacific. When it reached the general position of subtropical high, it turned back to the
direction of northwest before finally reaching the site. The high pressure gradient between the center
and edge of the subtropical high was opposite to its moving direction, which might be the cause of its
abrupt turning. Although the southwestern edge of the subtropical high itself had wind flows whose
directions were away from the Australia continent at a monthly average (Figure 13), the wind flows
from northern edge of southern hemisphere's westerlies could still move along its southwestern edge as
soon as they intersected with each other if smaller time scale and single level were considered (Figure
14b). Figure 14f showed the contributions of main backward trajectories based on the whole month's
statistics. It can be seen that the wind flows which could potentially bring aerosols from the fire center
still had a little contribution, which accounted for 9.32% (2.87%+6.45%). In contrast, winds coming
from the Pacific to the east and northeast of the Australia continent dominated the wind field at the site,
whose contributions were 25.09% and 54.12% respectively. Thus, the aerosol contribution from wind
transport should be limited, which was only found during one time period with time length less than 10
days in January 2020. From the perspective of multi-layers of atmosphere (0-3 km), the multi-layers of
background wind fields as a whole did not contribute to the aerosol and $CO_2$ transport from the fire
center to the site. Therefore, the soar of aerosols including BC and OC at the site should be mainly

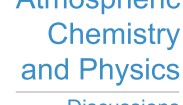

caused by the combined effect of combustion in the fire center and great free diffusion caused by
significant concentration gradient, with likely relatively weak contribution of the wind transport.
Most aerosols are generally within atmospheric boundary layer under normal conditions while it might
be different under the situation during mega fire events considering the boost of vertical movement due
to great heat release from fires and astonishing amounts of aerosol loadings. Smoke, as a kind of
unique aerosol loading with great amounts during fire events, could be essential to SW and LW speed
anomaly due to its absorptive radiative properties, making it particularly valuable to examine its
transport individually. Yang et al. (2021) analyzed the vertical distribution of smoke on southeastern
parts of Australia, which included the fire center and the site, and found that the smoke accumulated at
3 km generally. Considering this, we also did the backward trajectory analysis at 3 km whose time
division was the same as that at 500 m. The results are shown in Figure 15. As shown, the wind flow
scattered more evenly at 3 km than at 500 m. There were more wind flows coming from the
southwestern direction of the site. This is probably due to the fact that the magnitude and stretching
area of westerlies are larger at upper atmosphere than at layers closer to the surface. During period a, b
and e, there were clusters of wind flows coming from the fire center or near the fire center, which could
bring aerosols to the site. Specifically, there were wind flows penetrating the fire center directly during
period a and e, while the wind flows during period b are only adjacent to the north edge of fire center.
Since the period b was the longest among all No-SLB periods, it did not necessarily mean that the
wind's aerosol transport effect during this period was less than those during other periods although the
wind flows were not directly from the fire center. The moving paths of them were similar as that of
wind flows in Figure 14b, which all had an abrupt turning on the Pacific to the southeast of the site.
This is probably because that the south hemisphere's subtropical high developed to be quite strong
during the middle of summer, making the pressure gradient exist both at 500 m and 3 km. Figure 15f
shows the contribution of wind flows on monthly average, whose clustering number was also four.
There were four main directions of wind flows, whose contribution were 28.67%, 21.86%, 11.47% and
37.99% respectively. In order to make it clear, we define these four main wind flows as wind flow
clusters. The wind flow clusters with contributions of 21.86% and 11.47% were generally adjacent to
the north edge of the fire center, which contained contribution of wind flows from the fire center. Due
to the clustering limitation of Meteoinfo, we could not extract the specific contributions of wind flows
blowing directly from fire center from the total contributions of wind flow clusters (21.86% and



11.47%). But based on analysis on shorter time periods, their contributions were larger than those at
500 m because there were more No-SLB periods with wind flows blowing from fire center.
**4. Summary and discussion**
This In this study, the SLB day number, SLB speed, daytime temperature and nighttime temperature at
Brisbane Archerfield during January were calculated from 2001 to 2020 using observation data from
automatic meteorological station. We have taken three steps in total to exclude the interference of
winds from synoptic-scale systems in order to extract the real SLB signals. First, we used SRWF
method to verify the OE-SLB and then extracted the SLB signal from original observation. Second, we
defined SLB day when the whole SLB circulation is most significant and integrated. Finally, we used
SLB signals during PTS (PTL) on SLB days to calculate the monthly average of SW (LW) speed.
During the corresponding month over the years, regional cloud fraction, COD, fire spot and FRP
distribution in Australia were revealed using MODIS product. Aerosol distributions in eastern Australia
were revealed in the form of AOD using MERRA-2 product, including that of total aerosols, OC and
BC. Furthermore, the background wind field and backwards wind trajectory were analyzed by ERA5
product and HYSPLIT respectively. The main findings of this study are as follows.
1). There was a significant slump in SLB day number (33.3% of the average level) and LW speed
(decreased by 22.3% of the average level) at the site. While SW speed also decreased by 14.8% of the
average level, it was not significant.
2). There was a burst of aerosols at the site, with TA-AOD, BC-AOD and OC-AOD approximately
240%, 425%, 630% of the multi-years' averages. TDLS is the direct cause of SLB while other factors
influence SLB through their effects on TDLS. The variation of nighttime land temperature could
generally represent the variation of TDLS during nighttime while TDLS during daytime could not be
simply represented by daytime land temperature. Specifically, the significant aerosol burst was mainly
responsible for the decrease of SW speed. The burst of BC at the site as well as the large-scale SST
decrease during mega fires were mainly responsible for the slump of LW speed. $CO_2$ emitted by nearby
fire spots or transmitted from the fire center was a potential and weak factor for the slump of LW speed.
While the heating effect of nearby fires on TDLS was weak during both daytime and nighttime.
3). Emissions from fire center were mainly responsible for the local positive aerosol anomaly during



mega fire events. On average, the background wind fields from near surface to 3 km were not favorable
for aerosol and $CO_2$ transport. But there were likely aerosol and $CO_2$ transports through large scale
wind field at single level during shorter periods within January of 2020. Specifically, the wind flow
transport at 3 km was stronger than that at 500 m, which was particularly important for smoke transport
since the smoke from fires gathered at the same level. In general, free diffusion due to large
concentration gradient was mainly responsible for aerosol transport and the potential $CO_2$ transport
while the effect of background wind field played a second role.
In order to make it clear and concise to the influencing factors of SLB, we summarized their potential
mechanisms in local sea-land system (Figure 16). During daytime, negative anomaly of SW speed was
found at the site during January in 2020 when Australia mega fires were most intensive. The local
cloud fraction and COD were almost on an average level while there were much more aerosols during
mega fire events, which mainly came from fire center by free diffusion. They significantly weakened
the *in situ* downwelling solar radiation thus further narrowed the TDLS, which was the direct cause of
SW speed decrease. BC and $CO_2$ heated the atmosphere and warmed the earth-atmosphere system by
longwave radiation from the heated atmosphere. Warming effect of BC and $CO_2$, the decrease of SST at
a large scale and the weak heating effect of nearby fire spots partially offset the effect of aerosols on
narrowing TDLS, making the negative SW speed anomaly not exceed the multi-years' oscillation range.
During nighttime, the heating effect of nearby fire spots was still weak but more significant than that
during daytime. The warming effect of BC and $CO_2$ was like adding a heater in the atmosphere, which
triggered a SW circulation anomaly thus resulted in a slump in LW speed. The decrease of SST at a
large scale further boosted the decrease of LW speed. The slumps in both SLB speed and SLB day
number could help to accumulate the local aerosols (Shen and Zhao, 2020), which further catalyzed the
physical processes mentioned in the mechanism and finally formed a positive feedback mechanism
under a scenario of mega fires.
Essentially, narrowed TDLS was the direct cause of SLB speed decrease, which was affected by
various factors in the form of either shortwave radiation or longwave radiation. It not only weakened
the SLB speed, but also brought about a slump in SLB day number. The *in situ* radiation, including
both longwave and shortwave radiation reaching the ground, has a direct impact on the TDLS
considering the basic physical mechanism of SLB formation. Note that the specific weather condition,
cloud fraction, COD, and the type of clouds and aerosols could all affect the *in situ* radiation. Apart





from *in situ* radiation, the heat release in urban areas, heat waves, heating effect of nearby heat sources,
large-scale signals of SST and land surface temperature variation could all affect TDLS by changing
either the local land temperature or SST. The large-scale signals of temperature variations could be
caused by either natural variability or human variability. Normally, SLB forms when the TDLS is
obvious and the background wind field is mild. So the condition of large scale wind field such as
monsoon is also an important influencing factor of SLB. Apart from the slump in both SLB day
number and LW speed during mega fire events, there were fluctuations in both of their trends, which is
need further study in future.
***Data availability.*** The Dynamic Land Cover Dataset (DLCD) can be approached thorough Geoscience
Australia          (http://www.ga.gov.au/scientific-topics/earth-obs/accessing-satellite-imagery/landcover,
Lymburner et al., 2015). MERRA-2 Reanalysis data can be approached through the NASA Global
Modeling and Assimilation Office (https://gmao.gsfc.nasa.gov/reanalysis/MERRA-2/, GlobalModeling
and Assimilation Office (GMAO), 2015). MODIS observation data can be approached through
Earthdata center managed by NOAA (https://earthdata.nasa.gov/search?q=MCD06). GDAS data used
in HYSPLIT data are accessible through the NOAA READY website (http://www.ready.noaa.gov,
NOAA, 2016). Fire spot and FRP data can be approached from MODIS MCD14 product managed by
NOAA (https://earthdata.nasa.gov/search?q=MCD14). The wind and temperature observation data
from NOAA global observation network can be approached by NOAA's official website
(http://www1.ncdc.noaa.gov/pub/data/noaa/). The ERA5 data can be approached through official
website of Copernicus project (https://climate.copernicus.eu/climate-reanalysis).
***Author contributions.*** CFZ and LXS developed the ideas and designed the study. LXS, XCY, YKY and
PZ contributed to collection and analyses of data. LXS and XCY performed the analysis and prepared
the manuscript. CFZ supervised and modified the manuscript. All authors made substantial
contributions to this work.
***Competing interests.*** The authors declare that they have no conflict of interest.





**Acknowledgements.** This work was supported by the Ministry of Science and Technology of China

National Key Research and Development Program (2019YFA0606803), the National Natural Science

Foundation of China (41925022), the State Key Laboratory of Earth Surface Processes and Resources

Ecology, and the Fundamental Research Funds for the Central Universities.

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

**Figures and tables**

Table 1: Summary of information for the verification of OE-SLB at Brisbane Archerfield.

| The range of SW | The range of LW | PTS (UTC) | PTL (UTC) |
| --- | --- | --- | --- |
| [20° 135°] | [200° 315°] | [500 800] | [1400 2000] |


Table 2: Summary on the effect of different factors on TDLS. Factors marked in red represent that

they are either weak factor or potential factor derived from theoretical analysis but not verified by

observation.

| Influencing factors | | Forcing on Daytime TDLS | Forcing on Nighttime TDLS |
| --- | --- | --- | --- |
| Large scale forcing | Cooling of SST on a large scale (Hirsch and Koren, 2021) | + | - |
| Regional forcing | Heating effect of nearby | + | - |





|  | fire spots |  |
| --- | --- | --- |
| Total aerosols | - | × |
| BC | + | - |
| OC | - | × |
| CO$_2$ | + | - |


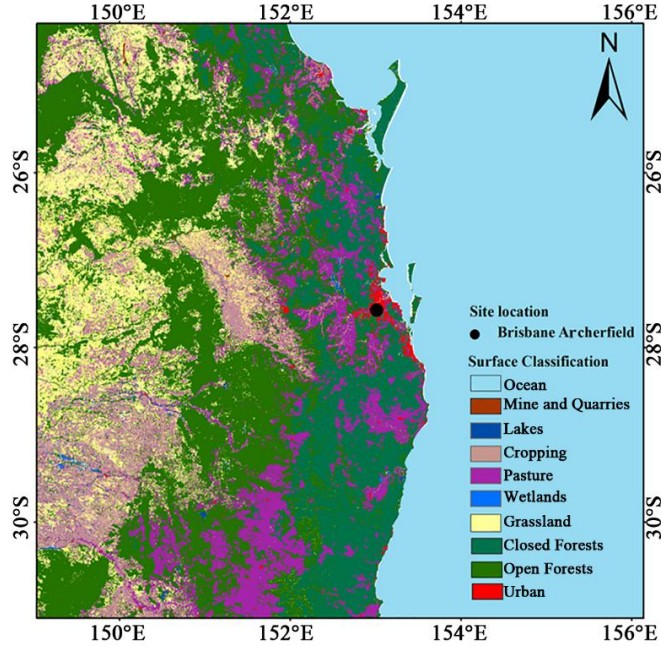


Figure 1: The map of eastern Australia with land-cover types. The observation site is marked in a black
dot.






Figure 2: Hourly average of wind angle in a diurnal period (HAWADP) of the local wind.

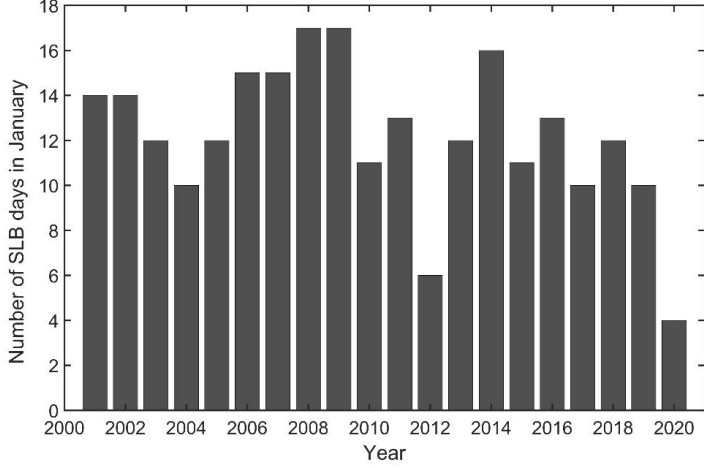


Figure 3: Number of SLB days in January from 2001 to 2020.



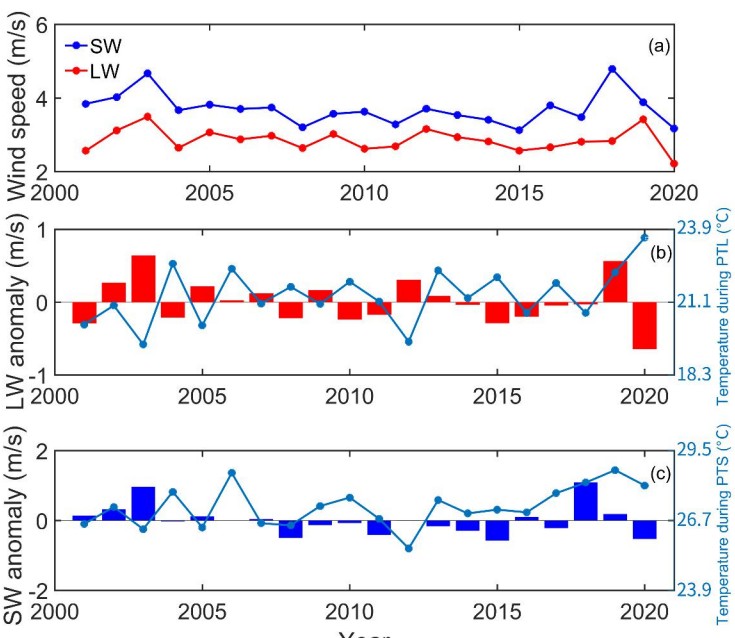

Figure 4: The trends of SW and LW speeds (a), the LW speed anomaly and land temperature during
nighttime (b), the SW speed anomaly and land temperature during daytime (c) based on the monthly
average of them during January from 2001 to 2020.

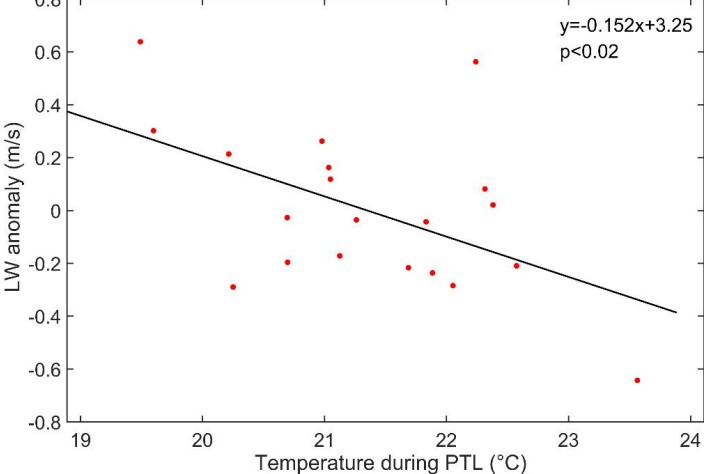

Figure 5: The relationship between LW anomaly and temperature during PTL based on monthly
average of them during January from 2001 to 2020.





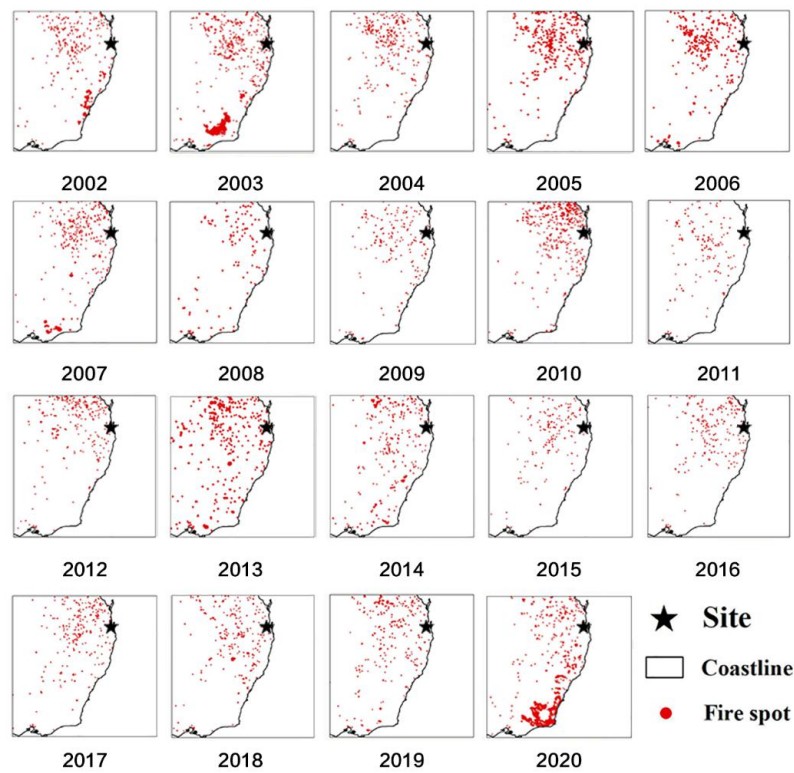


Figure 6: The fire spot distribution in the eastern Australia during January from 2002 to 2020.

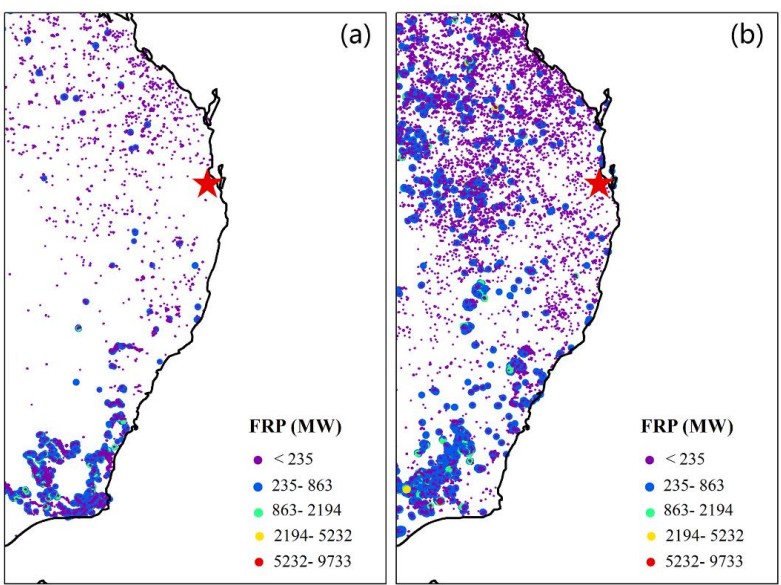


Figure 7: The fire radiative power (FRP) of total fire spots in eastern Australia during January in 2020



(a), January from 2002 to 2019 (b).

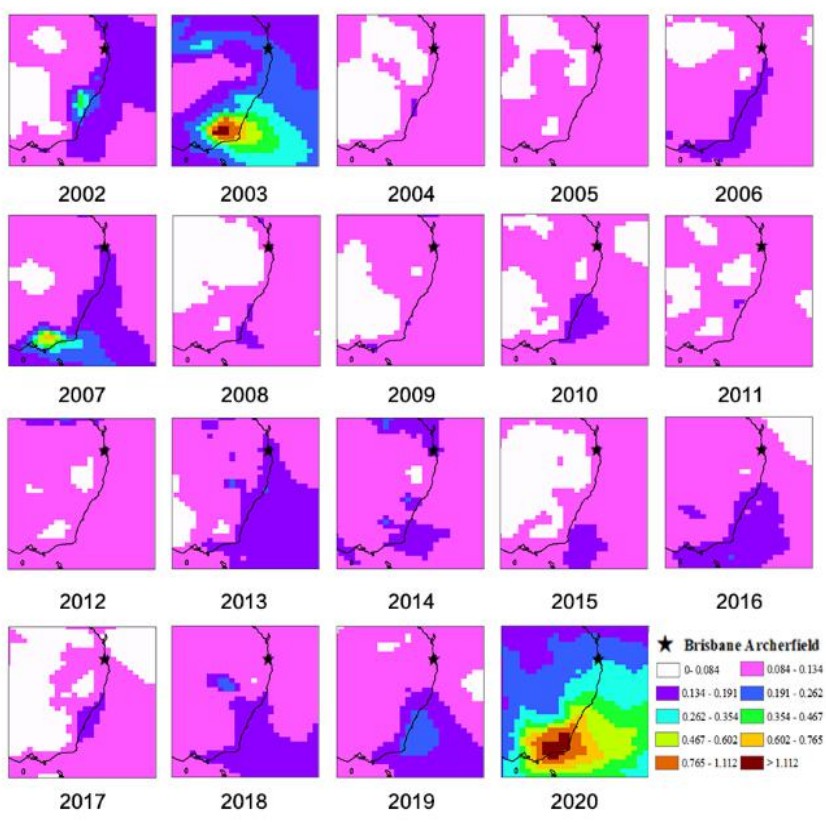


Figure 8: The spatial distribution of aerosol optical depth (AOD) of total aerosols in eastern Australia
during January from 2002 to 2020.



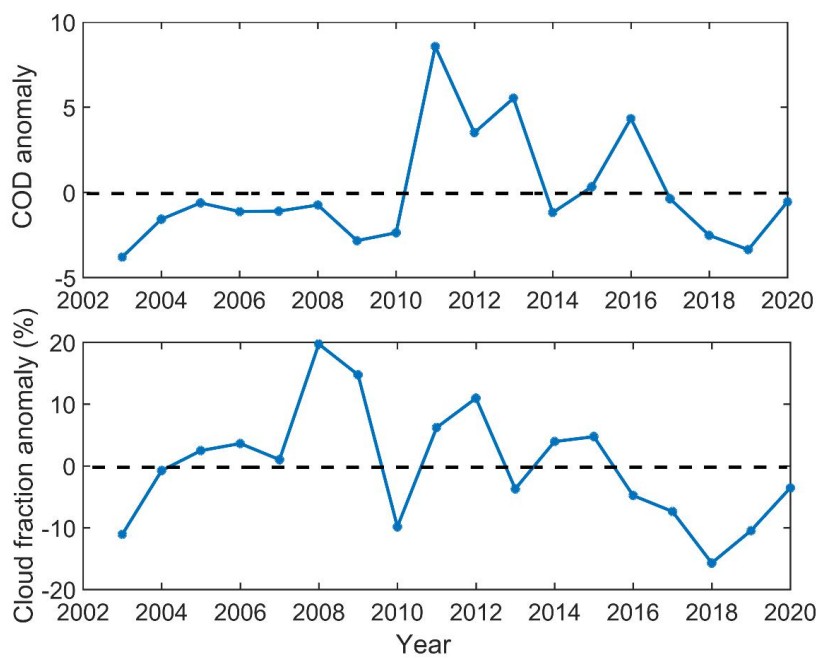


Figure 9: The monthly cloud optical depth (COD) anomaly and cloud fraction anomaly at Brisbane
Archerfield during January from 2003 to 2020.





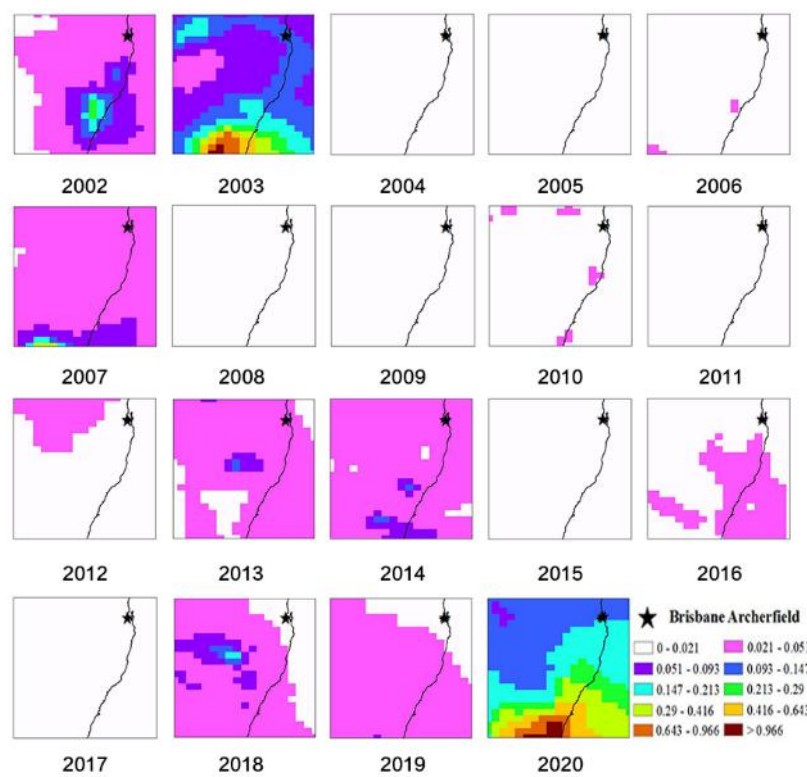


Figure 10: The spatial distribution of aerosol optical depth (AOD) of organic carbon (OC) in eastern

Australia during January from 2002 to 2020.



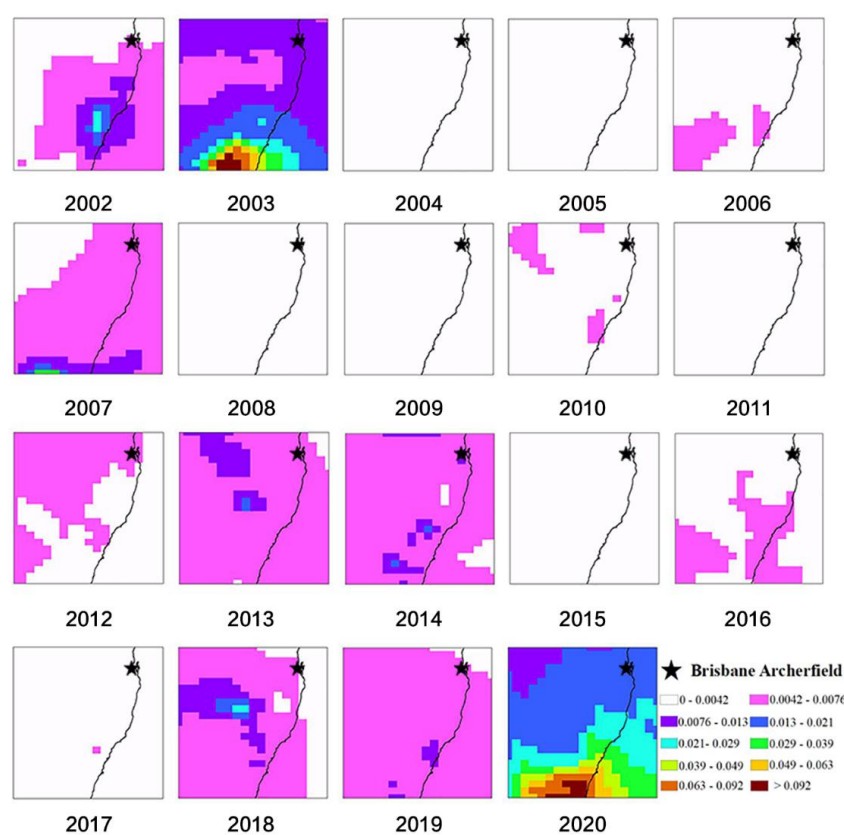


Figure 11: The spatial distribution of aerosol optical depth (AOD) of black carbon (BC) in eastern
Australia during January from 2002 to 2020.

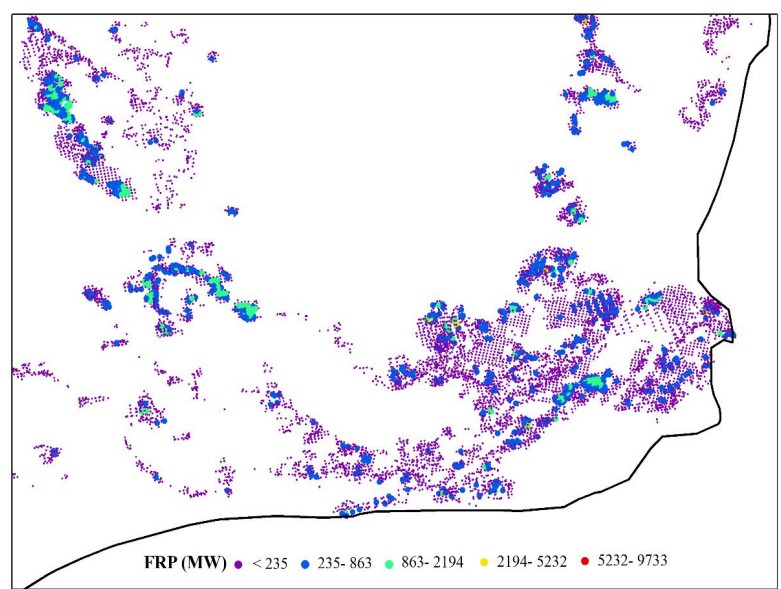

Figure 12: The detailed distribution of fire spots and their FRP in the fire center during January in 2020.

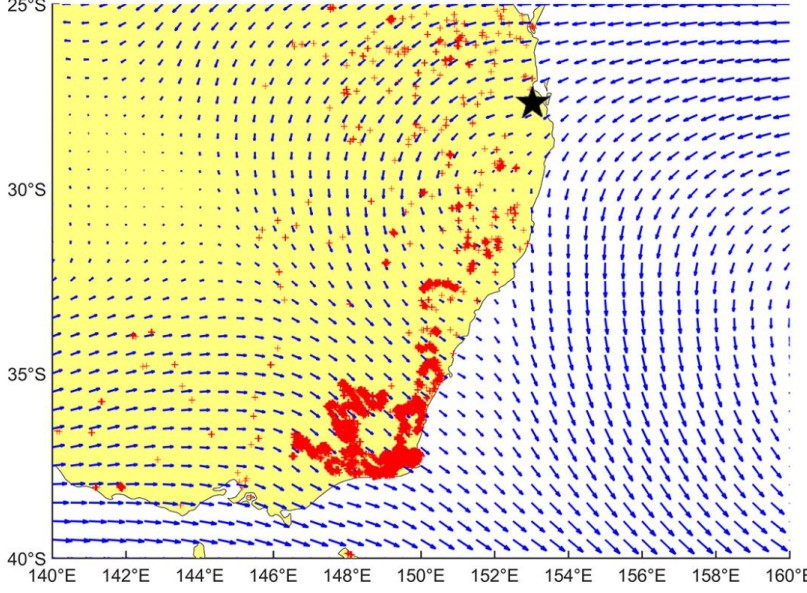

Figure 13: Monthly average background wind field based on wind information at pressure levels from 100hPa to 700hPa in January 2020. The red crosses present fire spots and the black star represents the



site location.

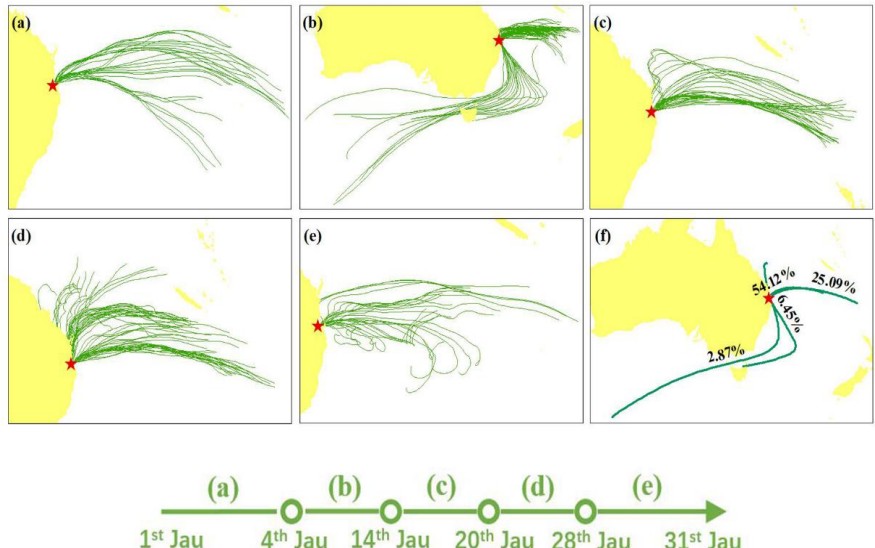


Figure 14: The site's the wind backward trajectories at 500 m during January in 2020. The wind
backward trajectories during first No-SLB period from 1st Jau to 3th Jau (a), the wind backward
trajectories during second No-SLB period from 5th Jau to 13th Jau (b), the wind backward trajectories
during third No-SLB period from 15th Jau to 19th Jau (c), the wind backward trajectories during fourth
No-SLB period from 21st Jau to 27th Jau (d), the wind backward trajectories during fifth No-SLB
period from 29th Jau to 31st Jau (e), the contribution of four main wind clusters based on the wind
backward trajectories during the whole month of January in 2020 (f).

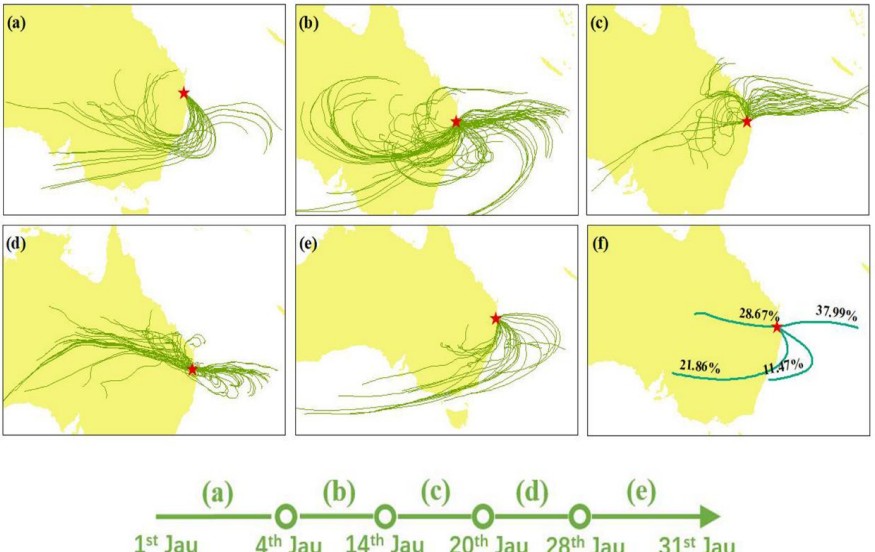


Figure 15: The site's the wind backward trajectories at 3 km during January in 2020. The wind

backward trajectories during first No-SLB period from 1st Jau to 3th Jau (a), the wind backward

trajectories during second No-SLB period from 5th Jau to 13th Jau (b), the wind backward trajectories

during third No-SLB period from 15th Jau to 19th Jau (c), the wind backward trajectories during fourth

No-SLB period from 21st Jau to 27th Jau (d), the wind backward trajectories during fifth No-SLB

period from 29th Jau to 31st Jau (e), the contribution of four main wind clusters based on the wind

backward trajectories during the whole month of January in 2020 (f).

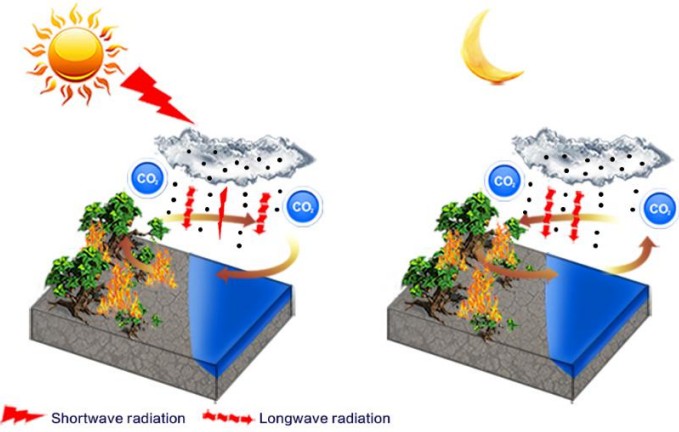

Figure 16: The summary of mechanisms containing influencing factors of local SLB during daytime





and nighttime. The black dots represent aerosols which include both scattering aerosols and absorptive

aerosols. The width of arrows of 'shortwave radiation' represents the magnitude of shortwave radiation.