# Peer review of "Observed slump of sea land breeze in Brisbane under the"

_Atmospheric Chemistry and Physics, 2021_

## Author Comment (AC1)

We sincerely thank the reviewers for their thoughtful, valuable and detailed comments and suggestions that have helped us improve the paper quality. Based on their comments, we have revised our manuscript carefully. Our detailed responses (Blue) to the reviewer's questions and comments (*Italic*) are listed below.

*Reviewer #1 (Comments to Author):*

*This study reported a reduction of sea land breeze (SLB), especially land wind speed associated with the great forest fire in Australia in January 2020. The author attributed the reduction to reduced surface downwelling solar radiation caused by increased aerosols, which serve to cool land surface more than the ocean, thus reducing land-ocean temperature contrast and wind speed. The conclusions are drawn through analysis and comparison of multiple observational and reanalysis products. While this find is plausible, it seems that there lacks a direct link between solar radiation and SLB strength. There are also some other minor issues that need to be clarified.*

We highly appreciate the reviewer's positive evaluation about the value of our study and invaluable comments. We made corresponding changes based on these comments as detailed below.

Major concern:

*All the conclusions are drawn based on the fact that surface downwelling solar radiation is directly linked to the SLB strength. However, there is no analysis of the change of surface downwelling solar radiation over both land and ocean, as well as the change of land-sea temperature contrast, during the fire episodes. These data should be available from surface sites, CERES satellite product, and reanalysis data, although the latter two are more uncertain. In any case, a direct investigation of surface solar radiation as well as temperature anomalies should help clarify the mechanism proposed.*

We highly appreciate the reviewer's valuable comments. Actually, the sea land breeze (SLB) system is a regional system whose range is about dozens of miles, though it generally varies depending on specific meteorological condition. According to its basic formation mechanism, its magnitude depends on the temperature difference between land and sea (TDLS). Note that TDLS means the temperature difference between the regional land surface and sea surface where the vertical streams of SLB circulation lie. These two surfaces do not necessarily have fixed locations but they depends on the specific regional temperature gradient distribution. This is quite a small scale and in reality there is no bouy recorder at the near-coastal sea surfaces. Even if we can put a bouy recorder there, it is not necessarily located at the surface where the vertical streams of SLB circulation lie. So it is impractical to detect the exact TDLS in this study. During nighttime, the sea surface temperature is comparatively stable. Consequently, the variation of land temperature can generally represent the variation of TDLS. We have detailed mathematical deduction supporting this idea in our previous study [Shen et al., 2021]. Theoretically, the land temperature anomaly should be a good indicator of LW anomaly, which is also supported by the

observation outcome (Fig. 5). Both heating effect of nearby fire spots whose effect was stronger than that during daytime and black carbon (BC)' warming effect contributed this narrowed TDLS, thus caused a slump of LW speed.

During daytime, the TDLS cannot be simply represented by daytime land temperature and it mainly depends on the in situ surface downwelling solar radiation (SDSR), though we still cannot detect the exact TDLS directly [Shen et al., 2021]. Unfortunately, there is no real in situ observation for SDSR. The in situ 'observation' of solar exposure is actually calculated fused with satellite data. As mentioned by Australia Weather Bureau, '**Clearly it would be impractical (not to mention exorbitantly expensive and labor intensive) to maintain high quality solar measurements at all locations across Australia. To circumvent this problem scientists (notably Dr. Gary Weymouth from the Bureau of Meteorology Research Centre) have developed a computer model using visible images from the geostationary meteorological satellites to estimate daily global solar exposures at ground level. To estimate the daily radiant exposure at each location, the images are averaged over at least four pixels and integrated over the entire day**'. (http://www.bom.gov.au/climate/austmaps/solar-radiation-glossary.shtml#globalexposure). Moreover, what the instrument detects are all shortwave signals reaching the surface. During mega fires, there are quite a lot diffuse solar radiation as well as diffuse shortwave radiation released by fire spots, bringing a lot of uncertainties to the detection of SDSR even if there was an instrument, which is the direct driver of sea wind (SW) as we mentioned. Considering all these aspects, we choose to investigate the distribution of SDSR using CERES. The outcome is shown as below (Fig. R1):

[Figure]

Fig. R1. The distribution of monthly mean surface downward shortwave radiation in January from 2001 to 2020 based on CERES data.

Fig. 4 shows that there were obvious negative SW anomalies in 2008, 2011, 2015 and 2020. As can be seen in the figure, the SDSR at the site was also at low level in these years. There were obvious positive SW anomalies in 2002, 2003 and 2018. The SDSR at the site (Fig. R1) was also at high level in these years. The increased SDSR in 2003 and 2018 may be caused by low level of cloud fraction and COD, whose influence should not be ignored in this proposed mechanism (Fig. 10 & **Lines 439-461**). We also note that there were some years when the radiation was high but the SW speed was not very high. For example, though lower than those in 2002 and 2003, there was high SDSR in 2019 but SW speed was only a little higher than normal. Note that CERES data is partially based on model simulation and its spatial resolution is coarse. The model simulation takes aerosols into consideration but it cannot accurately record the vertical distribution and different types of aerosols. Usually, it is suitable to use it to investigate the large-scale distribution of radiation or seasonal variation of radiation over a large area. However, it might bring uncertainties to the value of radiation in terms of regional scale. Just take 2020 as an example, the AOD at the fire center was more than 10 times than normal condition (Figs. 8-9), which should bring obvious changes to SDSR considering such an enormous release of absorbing aerosols. We do see that the SDSR at the fire center was at the low level in 2020, CERES generally reveals this phenomenon but it is similar as those in 2011, 2015 and 2016. In conclusion, CERES data have uncertainties to some extent, while the SDSR generally agrees well with the SW anomalies in Fig. 4.

In addition to potential observation support, we emphasized the physical mechanism of our analysis. The local cloud fraction and cloud optical depth (COD) were nearly at the average level, which ensures that the the increased absorbing and scattering effect brought by the aerosol burst would not be offset by significant cloud anomaly. Note that the exclusion of cloud's influence is important for the proposed mechanism. Consideration of this further makes our site as an ideal place to learn aerosols' effect on SLB. Importantly, the in situ SDSR is quite sensitive to the variation of AOD because of aerosol's direct effect [Turnock et al., 2015]. The AOD of total aerosol increased significantly during mega fires. Though we lack accurate in situ observation of SDSR due to several limitations as mentioned before, all of these ensure that the in situ SDSR should have negative anomaly and was linked to the SW decrease during mega fires from the aspect of physical mechanism.

Shen, L. X., Zhao, C. F., Yang, X. C.: Insight Into the Seasonal Variations of the Sea-Land Breeze in Los Angeles With Respect to the Effects of Solar Radiation and Climate Type, J. Geophys. Res.-Atmos., 126, 1-21, https://doi.org/10.1029/2019jd033197, 2021.

Turnock, S. T., Spracklen, D. V., Carslaw, K. S., Mann, G. W., Woodhouse, M. T., Forster, P. M., Haywood, J., Johnson, C. E., Dalvi, M., Bellouin, N., and Sanchez-Lorenzo, A.: Modelled and observed changes in aerosols and surface solar radiation over Europe between 1960 and 2009, Atmos. Chem. Phys., 15, 9477–9500, https://doi.org/10.5194/acp-15-9477-2015, 2015.

Minor points:

*The result based on one site does not seem robust enough, as there may be other*

*small-scale variabilities. Could the authors examine and compare several sites affected by the fire?*

We thank the reviewer for the valuable comment. Actually, the fire's center was located at the southeastern corner of Australia continent (Fig. 6-7). It should be the best if we could choose sites there for this study. But the sea land breeze (SLB) there is not significant due to typical climate type, as we have indicated in our previous study [Shen et al., 2021]. Moreover, severe fires can induce complex flows thus bring uncertainty for SLB identification. So we decide not to select sites there. We have clarified this in detail in **Lines 120-134**. Apart from the fire center areas, along the eastern coastal areas of Australia which are nearer the fire center than Brisbane, most of the areas are covered with forests. This brings two disadvantages for site selection. Firstly, many sites are rural sites which lack continuous observation with high time resolution. Secondly, due to the existence of photosynthesis, the heat absorption process of leaves from solar radiation and the temperature rise of 'leave surface' are different from those of Earth surface. As a result, the traditional mechanism of SLB formation is not necessarily applicable when the site is in the forest or quite close to the clusters of flora. Along the eastern coastal areas of Australia which are nearer the fire center than Brisbane, urban site like Sydney is not covered with flora but is too large in terms of urban scale, which could possibly bring urban heat island (UHI) circulation and uncertainties for SLB formation or identification. Coastal sites to the north of Brisbane are too far from fire center, and they are mostly rural sites covered with flora as well (Fig. 1). Considering all these aspects, Brisbane area is the most suitable for SLB study during mega fires. We have revised our manuscript and make a clearer clarification in **Lines 132-155**.

In order to make the conclusion more solid, we investigated another site over Brisbane area, whose name is 'Brisbane Airport' and located at '153.067°E, 27.417°S'. It is to the northeast of the Brisbane Archerfield, with long-term available observation with high time resolution (Fig. R2). We use the similar method for all steps until the final SLB speed calculation. The corresponding outcome is shown in Fig. R3. As can be seen in Fig. R3, the sea wind (SW) speed also had a significant negative anomaly in 2020, which is in accordance with what we have found at the original site. But unlike the outcome at the original site, the land wind (LW) did not show negative anomaly at Brisbane Airport (supplementary site). As can be seen in Fig. R2, most of the LW range at the supplementary site includes directions of the nearby mountains, which are covered with flora as well. Not only may the existence of flora surface bring about uncertainties to SLB formation and detection, but also there might be mountain-valley wind in the direction of LW range. Possibly, it can interrupt the detection of real LW signals. Moreover, the mega fires can also make a difference to mountain-valley wind itself, whose signals of variation might mix with those of LW signals. Both of these make the outcome of LW speed anomaly in Fig. R3 unreliable. Meanwhile, most of the LW range at the original site excludes directions of the nearby mountains, which is also one reason for the original site to become the typical site of SLB research during the mega fire. Apart from the reasons why we choose this site in Section 2.1, we have added the above reason why we selected this site in Section

2.3.1 when we first mentioned the range of LW (SW) at **Lines 232-235**. In any case, there is no difference between the areas in the direction of their SW range (20°-135°), making both outcomes of SW speed anomaly in Fig. R3 and Fig. 4 reliable. This can make our conclusions more robust.

[Figure]

Fig. R2 The location of the original site of Brisbane Archerfield (red marker) and the supplementary site of Brisbane Airport (blue marker). The orange lines show the range of land wind (LW). The geoscience information is based on Baidu online map system (https://map.baidu.com/).

[Figure]

Fig. R3 The trends of land wind (LW) and sea wind (SW) speed anomalies in January from 2000 to 2020.

*Section 2.2: Have the authors checked the quality of MERRA-2 AOD? Maybe a comparison with MODIS will help. Re-plotting Figure 8 with MODIS data is also recommended.*

We thank the reviewer for the valuable comment. We have added a figure (Fig. 9) using monthly MODIS AOD product. It can be seen that except for a little overestimation of AOD in the fire center in 2020, the overall distribution and value of AOD revealed by MERRA-2 agreed well with those revealed by MODIS. Especially for the site learned in this study, the difference in AODs between MERRA-2 (approximately 0.26) and MODIS (approximately 0.29) was very small. Both MERRA-2 and MODIS show that there was a burst of aerosols in the fire center during January in 2003 and 2020 and the latter was much more severe. Previous studies have also confirmed the quality of MERRA-2 AOD product by comparing with AERONET at several sites over coastal areas of East Australia which was also within the areas affected by the mega fire (**Lines 167-169**). So we believe that the quality of MERRA-2 AOD is ensured for this study.

[Figure]

Figure 9: The spatial distribution of aerosol optical depth (AOD) of total aerosols in eastern Australia during January from 2002 to 2020 using Moderate Resolution Imaging Spectroradiometer (MODIS) AOD product.

*Section 2.2: Ground observation also needs quality control. Moreover, I wonder if for every month the observation is complete, i.e., there are full hourly observations on*

*every day of the month? It may impact the SLB day statistics if there are missing observations on some days.*

The ground observation has undergone official quality control, with the help of combined efforts from the National Climatic Data Center (NCDC), Federal Climate Complex (FCC), the Unite State's Air Force and Navy etc. In order to exclude potential errors as thorough as possible, we selected an urban site. Usually, urban site is the flagship site over a large area and has served for longer time period, with better integrity of data and maintenance of instruments. Detailed information on quality control can be accessed online (https://www1.ncdc.noaa.gov/pub/data/noaa/ish-qc.pdf). The continuity of observation is ensured with observations on every 3 hours on each day in January from 2001 to 2020. Here, we have re-checked the continuity of observation. As shown in Fig. R4, the total number of sample days is 620 (31×20), with 596 days having full observations every 3 hours, with 22 days only missing one observation. So, we can make 2 conclusions based on the statistics: 1. There are no missing day in terms of observation. 2. There are full observations on almost every day, with very few missing data on few days. The time series could be generally thought as a time series with full observation. Since the SLB study here is based on climatological analysis, it will not affect the SLB day statistics. We have made corresponding revisions regarding the continuity of the observation data at Lines 193-196: "**The continuity of the observation data is ensured, there are observations on each day in January throughout the whole study period, with only one missing observation data at each day of a small fraction time (approximately 3.5%).**".

[Figure]

Fig. R4 The statistics on the number of sample days with different daily observation times.

*Figure 5 is important, could the authors provide the correlation coefficient? Could they also show LW vs. Downwelling solar radiation?*

Thanks for the comment. We have added the correlation coefficient in Fig. 5 and made corresponding revisions in our manuscript. The R value is about 0.52, which is at the medium level. Based on the theory of statistics, we cannot simply judge the correlation relationship between two variables according to correlation coefficient only. The significance level is 0.019 (<0.02) so it means that there does exist the correlation relationship between these two variables. In any case, the samples tend to have a higher significance level as the total number increases. But the samples here have passed the significance test even though the number was at a low level, which further enhanced the reliability of the conclusion that they are correlated with each other, though their correlation coefficient may be not so high because of low level of sample number.

Moreover, we have noticed there was an abnormal point in 2019. This point in Fig. 5 has deviated from the overall trend of these two variables. There might be other potential disturbance in this single year. For example, normally the variation of land temperature during nighttime can represent that of TDLS due to comparatively stable condition of sea surface temperature (SST). In this single year, land temperature during nighttime increased but LW anomaly did not decrease as the linear regression trend shows. There might also be an occasional increase of SST near the coast over the area where vertical stream of SLB lies. As a result, the LW anomaly maintained at a high level though the land temperature during nighttime increased (SST's variation kept the TDLS being in accordance with LW anomaly, causing the point deviating from the linear regression's trend). Fig. R1 shows that there was a significant increase of SDSR near the coastal sea of the site, which might be a possible cause for SST increase during nighttime in this single year. This analysis is not the focus of the study so we have added a concise interpretation at **Lines 328-335**. In any case, if we excluded this abnormal point and carried on the linear regression based on other normal points, the R value and significance level would be 0.69 and 0.0012 respectively. Considering all these aspects, we kept holding the conclusion that these two variables are correlated with each other. In addition, we have emphasized at **Lines 324-327** that similar conclusion has been found in our previous study [Shen et al., 2021], in which we have given a detailed analysis on physical mechanism as well as some mathematical deduction.

Actually, as we mentioned in our manuscript, LW circulation happens during nighttime. From the aspect of SLB formation mechanism, SDSR is not its direct driver, though occasionally it can be one of the possible influencing factors of coastal waters' SST where the vertical stream of SLB lies. Instead, we investigated the TDLS, serving as the driver of SLB circulation, whose variation would be generally represented by that of land temperature during nighttime. In the following sections, we further gave a detailed analysis on the causes of narrowed TDLS during nighttime in 2020. Of course, TDLS also serves as a direct driver of SLB during daytime. Since the variation of daytime land temperature cannot represent its variation, we try to investigate the problem from the aspect of solar radiation, which serves as the direct cause of TDLS during daytime.

Shen, L. X., Zhao, C. F., and Yang, X. C.: Insight Into the Seasonal Variations of the Sea-Land Breeze in Los Angeles With Respect to the Effects of Solar Radiation and Climate Type, J. Geophys. Res.-Atmos., 126, 1-21, https://doi.org/10.1029/2019jd033197, 2021.

*Figure 7b caption: does the right panel show combined FRP from 2002 to 2019? I think a direct comparison should be made between FRP in 2020 and multi-year average, rather than multi-year sum.*

Thanks for the comment. Actually, the right panel represents the whole fire spots from 2002 to 2019 showing the FRP of all the fire spots. Since the specific location of fire spots differed from year to year, it is not feasible to calculate the multi-year average. Because we have investigated the density of nearby fire spots over the years, what we want to emphasize in this figure is that there was no discrepancy between FRP of nearby or local fire spots in 2020 and that of nearby or local fire spots in other years. So in this figure, we should focus on the color of nearby fire spots rather than density. Based on what we have found in Figs. 6 and 7, we can draw a conclusion that the heating effect of nearby fire spots did exist in 2020, contributing to the increase of land temperature to some extent, but it was not likely the major cause of land temperature anomaly. The heating effect was generally weak.

*Line 464-465: This does not seem correct. BC should have a cooling effect on the surface rather than warming.*

Thanks for the valuable comment. Actually, the warming effect of BC is tremendous and it serves as the second most important component of global warming after $CO_2$ in terms of high positive forcing [Jacobson, 2001]. The mechanism could be understood like this: BC absorbs large amounts of solar radiation in daytime, warms the upper atmosphere at certain level. During nighttime, there should be no solar radiation, and the warmed atmosphere would give out more heat to lower levels of earth-atmosphere system (if we see the lower level of atmosphere and land as a whole) in the form of longwave radiation, which is based on the Stefan-Boltzmann law. Since the driver of SLB circulation can be both shortwave radiation and longwave radiation during daytime and nighttime respectively, this is like adding a 'heater' in the upper atmosphere, just like the sun heating the regional land-sea system during daytime, and it will trigger an abnormal SW circulation. Of course, we can also analyze this from the aspect of vertical temperature gradient. Detailed information has been shown at **Lines 486-502**.

We do acknowledge that BC absorbs large amount of solar radiation during daytime and weakens the downwelling solar radiation reaching the surface, so this might offset the warming effect and result in its cooling effect on the surface on the daily average (might also has uncertainties considering its strong warming effect). But this does not affect the positive contribution to the warming of the lower level of earth-atmosphere system during nighttime.

Jacobson, M. Z.: Strong radiative heating due to the mixing state of black carbon in atmospheric aerosols, Nature, 409, 695-697, 2001.

*Figure 13 does not seem quite useful.*

Thanks for the comment. Actually, we here want to emphasize that the conclusion on the transport of aerosols may vary as the time scale changes. This figure makes a great contrast with the following figures showing different periods of aerosol transport. Moreover, it shows the general position of south-hemisphere's sub-tropical high, which is used in the analysis on the following figures. Considering all these, we think it is useful and choose to keep this figure.

*Section 3.5.2 and Figures 14&15, I wonder how the clustering of back trajectories is done? Is it manually or by some mathematical methods?*

Thanks for your question. The steps of clustering of back trajectories are as follows: First, Python + HYSPLIT was used to generate trajectories during this month (Jau 2020). Second, the TrajStat module from Meteoinfo version2.4.1 was used to cluster the back trajectories (http://meteothink.org/docs/trajstat/cluster_cal.html). Either Euclidean distance or angle distance can be an option of clustering. In this study, we used the Euclidean distance method for clustering. Certain mathematical method like the calculation of Total spatial variation (TSV) was used to determine the class number of back trajectories. More detailed information can be accessed through the introduction of the TrajStat module online (http://meteothink.org/docs/trajstat/cluster_cal.html). In order to make it clear, we have added a concise introduction at **Lines 604-607**.

*Figure 16: This figure is great. But may also mark aerosols, or BC, OC, etc. Or this figure gives the impression that CO2 is the primary contributor to SLB slowing down. Also, since aerosols transported make a difference, the transport pathway should be marked on this figure.*

We appreciate the great suggestions. We have made corresponding revisions to ensure this figure more readable. We have marked BC, OC and other types of aerosols which all absorb and scatter the solar radiation during daytime. During nighttime, there is no solar radiation. The warming effect of BC mainly contributed to LW slowing down, so we only marked BC at the local site. But burning happened during both daytime and nighttime, so we marked all types of aerosols in the fire center. We have also added the transport pathway of aerosol, that is, free diffusion of aerosols from the fire center with a higher concentration to remote areas with a lower concentration. The weak influence of $CO_2$ have also been marked using dashed lines. The revised figure is shown as follow:

[Figure]

Figure 17: The summary of mechanisms containing influencing factors of local SLB during daytime and nighttime. The larger fire cluster represents the center of mega fires with a higher concentration of all types of aerosols. During Australia mega fires, aerosols were transported to the local site by means of free diffusion, which was caused by the great concentration gap of aerosols between fire center and the local site. The width of arrows of 'shortwave radiation' represents the magnitude of shortwave radiation.

*Some language still seems a bit awkward, with some typos. Please carefully proofread before resubmission.*

Thanks for the valuable comment. We have carefully proofread the whole manuscript to avoid typos. In addition to this, we have used professional language service to avoid being awkward.

---

## Author Comment (AC2)

We sincerely thank the reviewers for their thoughtful, valuable and detailed comments and suggestions that have helped us improve the paper quality. Based on their comments, we have revised our manuscript carefully. Our detailed responses (Blue) to the reviewer's questions and comments (*Italic*) are listed below.

*Reviewer #2 (Comments to Author):*

*The 2019 Australia mega fires have got great concerns by the science community. Sea land breeze (SLB) is a regional thermodynamic circulation closely related to coastal atmospheric environment yet few have looked into how it is influenced by different types of aerosols transported from either nearby or remote areas. By focusing on the 2019 Australia mega fire events, this study investigates this issue and found that SLB day number during the great fire month was only four, accounting for 33.3% of the multi-years' average. The land wind (LW) speed and sea wind (SW) speed also decreased by 22.3% and 14.8% compared with their averages respectively. Potential mechanisms how aerosols are transported here and affect the SLB through radiative cooling have been carefully analyzed and discussed. The findings are great contribution to the science community and definitely worthy for prompt publication after some necessary minor revisions.*

We highly appreciate the reviewer's positive evaluation about the value of our study and invaluable comments. We made corresponding changes based on these comments as detailed below.

Minor points:

*Could the author clarify the continuity of the observation data? Since the SLB day is selected based on certain rules, it seems to be necessary that the original data is continuous throughout the study period, then it is meaningful to compare each year's SLB day number.*

Thanks for the valuable comment. We have checked the continuity of the data, including both the number of observation day and that of daily observation time. We now add the information at **Lines 193-196. 'The continuity of the observation data is ensured, there are observations on each day in January throughout the whole study period, with only one missing observation data at each day of a small fraction time (approximately 3.5%).'** The original time series are thought as generally continuous and are definitely suitable for SLB study. The total number of sample day was 620 (31*20), which means that there were observation records on each day of January from 2001 to 2020. As shown in Figure R1, the full daily observation time was eight with few days missing only one time of observation.

[Figure]

Fig. R1 The statistics on the number of sample days with different daily observation times.

*Line 43-44, "There were" should be "There are"*

Thanks for your careful proofreading. We have corrected it.

*Line 50-51, "during different seasons" is suggested as "in different seasons"*

Modified as suggested.

*Line 60, "is" should be "was"*

Corrected.

*Line 83, "Shen et al., 2021; Shen et al., 2021;" should be "Shen et al., 2021a, b"*

Corrected.

*Line 155, remove "The" for "The Several types of data"*

Corrected.

*Line 164: Please modify the decimal number of the spatial resolution to keep them uniform in terms of resolution.*

Thanks for the suggestion and we have made corresponding revision. Since the original version was '0.002349°×0.002349°', it is only appropriate to modify it to be '0.002°×0.002°'. We have also modified other spatial resolution to keep the decimal number uniform.

*Line 159, please confirm it is GADS or GDAS?*

It should be GDAS and we have corrected all wrong abbreviations.

*Line 200-204, please confirm and make it consistent for the dataset name, GDAS, GADS, or GDADS?*

Corrected.

*Line 243-244, "time period" is suggested as "period"*

Modified.

*Line 330-331, "which is the direct cause of SW speed decrease" should be "which is the direct cause of decreased SW speed".*

Corected.

*Line 388-390, considering the importance of solar radiation, it is worthy for the authors to check the change of radiation there. If there are ground-based observation of radiation, that will be great. If there are not, the authors might check the CERES radiation data while it might have too coarse spatial resolution to make the analysis not possible or challenging. Anyway, it is worthy to check if there are suitable data and check if the radiation has the expected changes. Of course, even if there are no radiation data, that would not affect the analysis here by directly considering the variation of temperature and aerosols.*

Thanks for the valuable comment. Unfortunately, there is no in situ observation at the site, cited by official stuff from Australia Weather Bureau, **'Clearly it would be impractical (not to mention exorbitantly expensive and labor intensive) to maintain high quality solar measurements at all locations across Australia. To circumvent this problem scientists (notably Dr. Gary Weymouth from the Bureau of Meteorology Research Centre) have developed a computer model using visible images from the geostationary meteorological satellites to estimate daily global solar exposures at ground level. To estimate the daily radiant exposure at each location, the images are averaged over at least four pixels and integrated over the entire day'**. (http://www.bom.gov.au/climate/austmaps/solar-radiation-glossary.shtml#globalexpos ure). Considering the lack of in situ observation, we choose to use CERES data to investigate the distribution of SDSR. The outcome is shown in Fig. R2.

Fig. 4 shows that there were obvious negative SW anomalies in 2008, 2011, 2015 and 2020, consistent with low levels of SDSR in these years at the site (Fig. R2). There were obvious positive SW anomalies in 2002, 2003 and 2018, with also high levels of SDSR at the site (Fig. R2). The increased SDSR in 2003 and 2018 may be caused by low level of cloud fraction and COD, whose influence should not be ignored in this proposed mechanism (**Fig. 10 & Lines 439-461**). We also note that there were some years when the radiation was high but the SW speed was not very high. For example, there was high SDSR in 2019 but SW speed was only a little higher than normal. Note that CERES data is partially based on model simulation and its spatial resolution is coarse. The model simulation takes aerosols into consideration but it cannot accurately record the vertical distribution and different types of aerosols. Usually, it is suitable to use it to investigate the large-scale distribution of radiation or seasonal variation of radiation over a large area. However, it might bring uncertainties to the radiation in terms of regional scale. Just take 2020 as an example, the AOD at the fire center was over 10 times than normal condition (Figs. 8-9), which should bring obvious changes to SDSR considering such an enormous release of absorbing aerosols. We do see that the SDSR at the fire center was at the low level in 2020. CERES generally reveals this phenomenon but it is similar as those in 2011, 2015 and 2016. In conclusion, CERES data have uncertainties to some extent, while the SDSR generally agrees well with the SW anomalies in Fig. 4.

In addition to potential observation support, we emphasized the physical mechanism of our analysis. The local cloud fraction and cloud optical depth (COD) were nearly at the average level, which ensures that the the increased absorbing and scattering effect brought by the aerosol burst would not be offset by significant cloud anomaly. Note that the exclusion of cloud's influence is important for the proposed mechanism. Consideration of this further makes our site as an ideal place to learn aerosols' effect on SLB. Importantly, the in situ SDSR is quite sensitive to the variation of AOD because of aerosol's direct effect [Turnock et al., 2015]. The AOD of total aerosol increased significantly during mega fires. Though we lack accurate in situ observation of SDSR due to several limitations as mentioned before, all of these ensure that the in situ SDSR should have negative anomaly and was linked to the SW decrease during mega fires from the aspect of physical mechanism.

Shen, L. X., Zhao, C. F., Yang, X. C.: Insight Into the Seasonal Variations of the Sea-Land Breeze in Los Angeles With Respect to the Effects of Solar Radiation and Climate Type, J. Geophys. Res.-Atmos., 126, 1-21, https://doi.org/10.1029/2019jd033197, 2021.

Turnock, S. T., Spracklen, D. V., Carslaw, K. S., Mann, G. W., Woodhouse, M. T., Forster, P. M., Haywood, J., Johnson, C. E., Dalvi, M., Bellouin, N., and Sanchez-Lorenzo, A.: Modelled and observed changes in aerosols and surface solar radiation over Europe between 1960 and 2009, Atmos. Chem. Phys., 15, 9477–9500, https://doi.org/10.5194/acp-15-9477-2015, 2015.

[Figure]

Fig. R2. The distribution of monthly mean surface downward shortwave radiation in January from 2001 to 2020 based on CERES data.

*Line 619, remove "This" in "This In this study"*

Corrected.